# Structural evolution during inverse vulcanization

Botuo Zheng [1,4], Liling Zhong[1,4], Xiaoxiao Wang[1], Peiyao Lin[1], Zezhou Yang[1], Tianwen Bai [2]✉, Hang Shen [3]✉ & Huagui Zhang [1]✉

Inverse vulcanization exploits $S_8$ to synthesize polysulfides. However, evolution of products and its mechanism during inverse vulcanization remains elusive. Herein, inverse vulcanization curves are obtained to describe the inverse vulcanization process in terms of three stages: induction, curing and over-cure. The typical curves exhibit a moduli increment before declining or plateauing, reflecting the process of polysulfide network formation and loosing depending on monomers. For aromatic alkenes, in the over-cure, the crosslinked polysulfide evolves significantly into a sparse network with accelerated relaxation, due to the degradation of alkenyl moieties into thiocarbonyls. The inverse vulcanization product of olefins degrades slowly with fluctuated relaxation time and modulus because of the generation of thiophene moieties, while the inverse vulcanization curve of dicyclopentadiene has a plateau following curing stage. Confirmed by calculations, the mechanisms reveal the alkenyl groups react spontaneously into thiocarbonyls or thiophenes via similar sulfur-substituted alkenyl intermediates but with different energy barriers.

Elemental sulfur, a major byproduct in oil refineries, has become a growing environmental concern due to its large production and limited application[1]. To address the issue, in 2013, J. Pyun and his coworkers[2] reported the copolymerization of elemental sulfur with multi-functional alkenes to synthesize polysulfides, named inverse vulcanization. The inverse vulcanization not only consumes elemental sulfur but also prepares dynamically crosslinked sulfur-rich polymers, which can be processed and repaired under heat via reversible cleavage and regeneration of S-S bonds[3]. Thanks to the excellent atom efficiency, low cost, and versatile sulfur-rich products, inverse vulcanization has been intensively exploited and investigated for various applications in the last ten years. A lot of techniques and systems have been developed to realize the inverse vulcanization of various monomers in mild conditions[4–9], and a broad scope of comonomers has been adopted[10,11] which gave rise to assorted polysulfides utilized in ion adsorption, Li-S batteries[12,13], optical materials[14,15] and other fields[16–18].

In great contrast to the prosperity in the applications and methodologies, a systematic illustration of inverse vulcanization mechanisms and relation between the chemical structures and reaction conditions is struggling. The problems lie in the fact that the crosslinking nature of polysulfides generated from inverse vulcanization of multi-functional alkenes hindered the dissolution of polysulfides and, thereby, their precise characterization by any solution techniques. Possible resolutions to facilitate the characterization include using the monofunctional alkenes as model comonomer[7,19,20] and the cleavage of the S-S bonds by reagents such as reductants[21] and nucleophiles[22]. J. Pyun et al. synthesized poly(styrene-sulfur) as a model polymer through inverse vulcanization of styrene for structure and mechanism investigation[20]. By inspecting the model polymers with nuclear magnet resonance (NMR) spectra, they successfully identified the linear, branched, and terminal units in polysulfides[20]. In other work, Y. Onose et al. carried out the inverse vulcanization of several aromatic and

[1]College of Chemistry and Materials Science, Fujian Key Laboratory of Polymer Materials, Fujian Normal University, Fuzhou 350007, China. [2]Key Laboratory of Medical Electronics and Digital Health of Zhejiang Province in Jiaxing University, College of Biological, Chemical Sciences and Engineering, Jiaxing University, Jiaxing 314001, China. [3]College of Materials and Chemical Engineering, Minjiang University, Fuzhou 350108, China. [4]These authors contributed equally: Botuo Zheng, Liling Zhong. ✉e-mail: baitw@zjxu.edu.cn; hang.shen@outlook.com; huagui.zhang@fjnu.edu.cn

aliphatic terminal olefins and reported a class of by-products containing 1,2-dithiol-3-thione moieties[23]. Some non-destructive techniques were also exploited to characterize the inverse vulcanization product. Raman spectroscopy[24] and X-ray photoelectron spectroscopy (XPS)[15,25] were demonstrated to be useful techniques in analyzing the structures of sulfur chain segments in the polysulfides. By deconvoluting the S $2p$ signals in XPS spectra and S-S vibration signals in Raman spectra, scientists can estimate the sulfur rank, i.e., average sulfur chain length between organic crosslinkers, and even distribution of sulfur chain length. In recent work, J. Pyun and J. Njardarson[21] used solid-state NMR to characterize the products generated from inverse vulcanization of 1,3-diisopropenylbenzene (DIB) and revealed the majority of bis-thiocumyl units in the poly(DIB-$r$-sulfur)s, which was confirmed by the structure analysis of polymer segments generated from the reduction of the polymer by LiAlH$_4$. Combining the experimental results with density functional calculation (DFT) results, they raised a systematic mechanism describing the reaction involving DIB and sulfur radical towards the bis-thiocumyl structure. In another study, J. Pyun and M. Mackey revealed the rheological properties of inverse vulcanized polymers in terms of their Maxwellian behaviors, which is exploited to infer the structure-property relationship of inverse vulcanization products synthesized from varied crosslinkers[26]. On the other hand, T. Hasell and his coworkers[27,28] did a series of work investigating the existence of the 'dark sulfur', i.e., unpolymerized amorphous polymer, in inverse vulcanization products and its roles in aging and recovery of the products. Thanks to the efforts from the research, a complete understanding of inverse vulcanization mechanisms and product structures is coming to light.

However, although there is a lot of research focusing on determining the properties and unit structures of inverse vulcanization products prepared from given conditions assuming a fixed structure, the evolution of the polysulfide network and segment structures during inverse vulcanization, as well as its mechanism, is still elusive. Among rare reports, J. Griebel found that in the inverse vulcanization of DIB, a too-long reaction time at high temperatures would lead to a flow transition of the polymer product, suggesting the breaking of the polymer network[29]. In contrast, D. Kim and his coworkers[30] discovered that the post-thermal treatment at 110 °C and 140 °C will improve the thermomechanical properties of inverse vulcanization products based on divinylbenzene in terms of glass transition temperature ($T_g$) and storage modulus. C. Jenkins et al.[31] investigated the effects of different reaction conditions, including reaction time, temperatures, scales, and feeding and heating procedures. The results showed that all the factors would influence the gel fractions, the glass transition temperature, and the molecular weight of soluble fractions, which are closely related to the degree of cross-linking. The complex relationship between reaction conditions and product properties observed clearly indicates that the reaction conditions and procedures adopted in different research greatly influence the structures and structural evolution of inverse vulcanization products, which dictates the network and segment behaviors of ultimate products. Nevertheless, only preliminary explanations were provided to elucidate the evolution of product structures and behaviors. The degradation of polysulfides at long reaction times was usually intuitively ascribed to the break and rearrangement of sulfur chains[28,30]. Moreover, most mechanism investigation and thermal treatment investigations were carried out with sole model monomer[20,21]. It is not sure whether the mechanism can be extended to other monomers, and the monomer dependence of the structural evolution is also obscure. For the reason of the lack of knowledge about the structural and property evolution as well as its relation with polymerization conditions and the alkene structures, the precise synthesis of polysulfides with controlled structures by inverse vulcanization is still challenging, which perturbs the development of polysulfide materials. Therefore, it is of great necessity to understand the evolution of inverse vulcanization products and the mechanism

behind them based on diverse alkenyl comonomers in various synthesis conditions.

In this work, we carry out a systematic study to unveil the structural evolution of polymer products during inverse vulcanization through small amplitude oscillation shear (SAOS) experiments and provide a detailed explanation of the mechanism behind the product evolution based on inverse vulcanization of multiple model monomers with different alkenyl functions. It was found that according to the in situ time sweep results, the inverse vulcanization exhibits three typical stages: induction, curing, and over-cure, similar to the vulcanization curve of rubber[32]. The frequency sweep followed by time-temperature superposition (TTS) and the relaxation spectrum established thereafter revealed that the products in every stage show different molecular weights between crosslinking or entanglement points, dynamic crosslinking behaviors and segment behaviors, while crosslinkers with different structures display distinct inverse vulcanization curves. The stages are also confirmed by the model inverse vulcanization of mono-function alkenes. The formation of the thiocarbonyl domains and their derivatives is demonstrated to be the major side reaction resulting in the degradation of the network and evolution of vinyl or isopropenyl monomers, especially when the alkenyl group is conjugated with aromatic rings. On the other hand, in inverse vulcanization of monomers with olefin chains, the polysulfide products degrade into thiophenes at relatively slow rates. The transformation mechanism into thiocarbonyl species and thiophenes depending on the monomer structures was verified to be favorable according to DFT calculations.

## Results and discussion

### Inverse vulcanization curves of crosslinker monomers

To study the evolution of inverse vulcanization products, a series of multi-function alkenes are chosen as the model crosslinkers, including DIB, 1,2-divinylbenzene (DVB), soybean oil (SO), and dicyclopentadiene (DCPD) (Fig. 1). DIB and DVB are the most extensively investigated crosslinkers for research into inverse vulcanization methods and mechanisms. Therefore, DIB and DVB are adopted in this work as representative monomers for aromatically conjugated alkenes[2,5,31], which is also readily compared with existing research results. In contrast, soybean oil, a typical bio-derived crosslinker, represents a large class of triglyceride monomers consisting of olefin chains, which was widely used to prepare processable and self-repairable materials by inverse vulcanization[33–35]. DCPD is also adopted as the model crosslinker because of its special cyclic alkene structure[36] and its role in strengthening inverse vulcanization products when copolymerized with other crosslinkers[37]. To avoid the dark sulfur[28] that may perturb the characterization and subsequent analysis, inverse vulcanizations of all the crosslinkers are studied at relatively low sulfur feed ratios (sulfur/alkenyl groups = 2–4) except soybean oil (sulfur/alkenyl groups = 4–6) which has sparse alkenyl moieties to sequester sulfur. On the other hand, it has been demonstrated that the reaction scale significantly influences reaction behaviors and the properties of the inverse vulcanization products[31] especially when the amount of feedstocks is large, because the high viscosity of the reaction mixtures in curing retards transfer of substances and heat released by exothermic polymerization. To exclude the disturbance of scale effect complicating the structural evolution, all the inverse vulcanizations were carried out with the same total weight (2.0 g) of feedstocks.

Inverse vulcanization of DIB is firstly investigated with a sulfur feed ratio of 2, 3, and 4 at varied temperatures (160, 170, and 180 °C, Supplementary Table 1). To monitor the evolution of the products during inverse vulcanization, the reaction mixture consisting of feedstocks and propagation species was subjected to a time sweep test in the linear viscoelastic (LVE) regime. To ensure the homogeneity of the reaction mixture during the rheological test, the elemental sulfur and the DIB with a small sulfur/alkene feed ratio of 2 were mixed and firstly

heated at 160 °C in a vial under stirring until a brown homogeneous mixture was obtained. Then the mixture was transferred to a pre-heated parallel plate in the rheometer and subjected to continuous SAOS tests at the temperature of inverse vulcanization (160 °C) (Fig. 1) to track the evolution of the products.

The storage modulus ($G'$) and loss modulus ($G''$) are plotted against time in Fig. 2a. The horizontal axis starts at the time when the sample was transferred to the rheometer instead of 0 min. In the inverse vulcanization of DIB at a sulfur feed ratio of 2 and 160 °C (DIBS1, Fig. 2a), at the beginning, the sample exhibited very weak viscoelasticity, even unable to afford enough signal-to-noise ratio for discernible signals. As the reaction proceeded at ~ 30 min, the $G'$ and $G''$ were detected to increase rapidly. After a maximum $G'$ ($G'_{max}$) was reached at ~70 min, both the $G'$ and $G''$ started to decline. Finally, a modulus plateau was approached after 250 min. The trend of the $G'$-$t$ curve clearly revealed that the physical properties of the inverse

vulcanization products were evolving during the reaction, which was also observed in recent literature[30]. In typical vulcanization of rubber by sulfur, the vulcanization process monitored by a rheometer could be divided into three stages: induction, curing, and over-cure stages in sequence (Supplementary Fig. 1), and in the over-cure stage the loss of the torque is usually observed[32]. Similarly, the inverse vulcanization process can be divided into three stages according to the $G'$-$t$ curve obtained from the time sweep test (colored zones in Fig. 2a), and the modulus evolution could be ascribed to the development of poly-sulfide network followed by its degradation analogous to the vulcanization of rubbers and the cure of resins. At the beginning of the test, the sample behaved as a dilute liquid under shear with small moduli lower than the detection limit of the rheometer ($10^{-6}$ MPa). The poor signals suggest that in this stage (gray zone in Fig. 2a), the mixture is majorly composed of oligomers with low molecular weights instead of crosslinked networks. This stage can be denoted as an induction

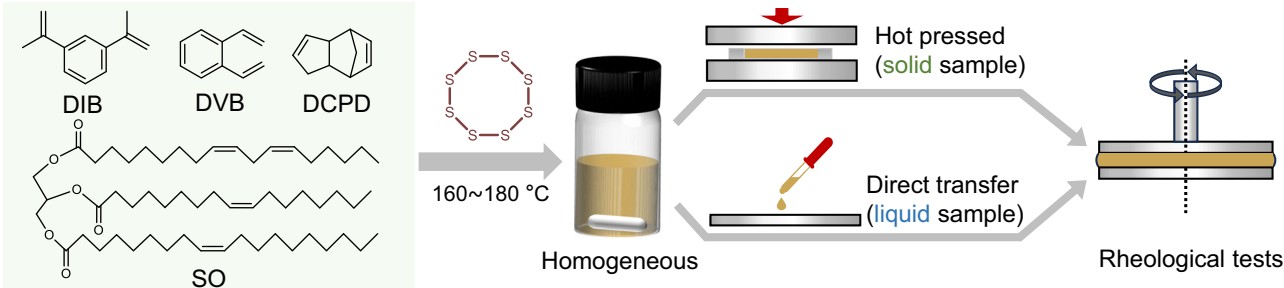

**Fig. 1 | Structures of the crosslinker monomers investigated and the experimental procedure of in situ time sweep test.** The monomer mixed with elemental sulfur was heated and stirred at inverse vulcanization temperatures into a homogenous liquid (DIB, DPCD, and SO) or solid (DVB). The liquid samples and hot-pressed solid samples were then transferred to pre-heated parallel plates for tests.

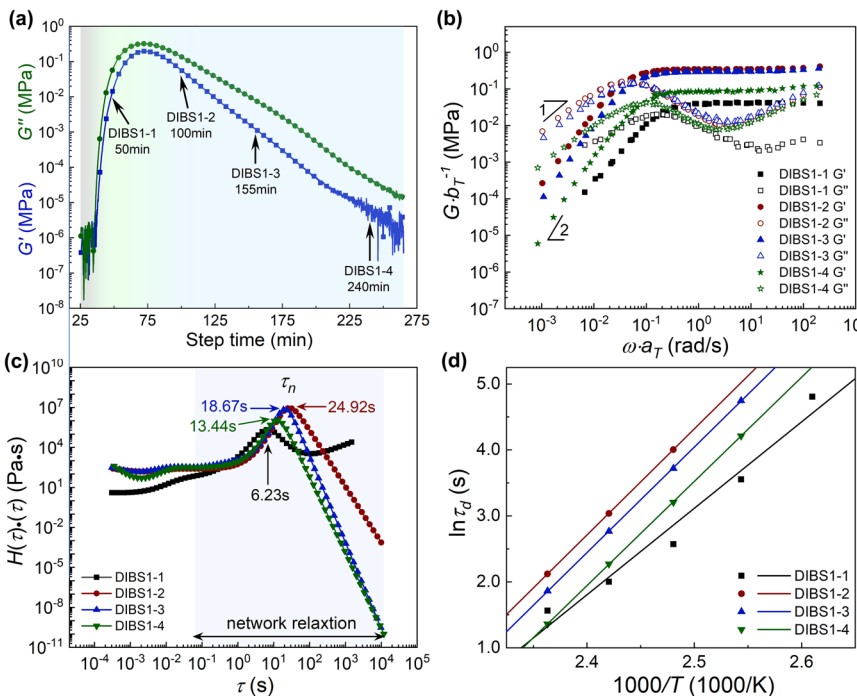

**Fig. 2 | Rheological test results of DIB inverse vulcanization. a** The time sweep test of the DIBS1 sample at 160 °C. The curve is shaded by three stages: to induction (gray), curing (green), and over-cure (blue) stages. **b** The frequency sweep results of DIBS1 samples at different inverse vulcanization times. The master curves are established from G·ω data at varied temperatures via the temperature-time superposition principle by each sample. **c** Relaxation time spectra of DIBS1 samples at different inverse vulcanization times. **d** Fitting of the characteristic relaxation time $\tau_d$ of DIBS1 samples at different inverse vulcanization times to an Arrhenius-type equation.

**Table 1 | Rheological test and characterization results of DIBS1 and SOS4 samples at different reaction times**

| Sample[a] | Reaction time (min) | $G_N^0$ (MPa) | $\rho$ (g/cm³) | $M_c$ (kDa) | $T_g$ (°C) | Gel fraction (%)[b] | $\tau_n$ (s) | $E_a$ (kJ/mol) | lnA | Sulfur rank[c] |
|---|---|---|---|---|---|---|---|---|---|---|
| DIBS1-1 | 50 | 0.0446 | 1.272 | 93.2 | 10 | 60 | 6.23 | 108.9 | −28.44 | 6.70 |
| DIBS1-2 | 100 | 0.322 | 1.324 | 13.4 | 35 | 90 | 24.92 | 133.7 | −33.46 | 5.96 |
| DIBS1-3 | 155 | 0.293 | 1.325 | 14.8 | 39 | 87 | 18.67 | 132.9 | −33.96 | 6.60 |
| DIBS1-4 | 240 | 0.0847 | 1.241 | 47.9 | 36 | 75 | 13.44 | 131.3 | −35.35 | 6.68 |
| SOS4-1 | 35 | 0.00170 | 1.067 | 237 | −12 | 0 | 64.07 | 110.7 | −29.65 | 7.24 |
| SOS4-2 | 75 | 0.0120 | 1.058 | 33.3 | −7 | 0 | 49.08 | 127.9 | −35.87 | 6.00 |
| SOS4-3 | 130 | 0.0230 | 1.078 | 17.7 | −9 | 0 | 46.24 | 129.5 | −35.94 | 6.56 |
| SOS4-4 | 240 | 0.0157 | 1.106 | 26.6 | −8 | 0 | 49.12 | 133.8 | −35.96 | 6.32 |

[a]The samples were subjected to frequency sweep tests from low to high temperatures (110–160 °C at an interval of 10 °C) at a strain amplitude of 0.2%, and the superposition curves was established afterward. [b]Gel fraction was measured by immersing samples in THF. [c]As determined by S 2p signals in XPS spectra.

period akin to the induction/scorch period of classical rubber vulcanization, wherein not yet or partially crosslinked rubber polymer chains still can be readily processed at given temperatures before the curing stage begins, as shown in a typical vulcanization curve (Supplementary Fig. 1). Following the induction period, the system moved into a curing stage (green zone in Fig. 2a), in which the alkene groups in the crosslinker were further consumed by radical species to form crosslinks. Therefore, a rapid increase in modulus can be observed until a max modulus is reached (~0.2 MPa). It is worth noting that the gelation or vitrification of the inverse vulcanization system in the vial visually occurred (54 min) before the max modulus was reached (80 min), suggesting incomplete inverse vulcanization when the vitrification is observed[30]. Subsequently, the inverse vulcanization entered an over-cure stage (blue zone in Fig. 2a) where the moduli of the sample significantly declined by orders of magnitude due to the proposed deterioration of samples at high temperatures. It is noted that $G'$ is always smaller than $G''$ throughout the curve (Fig. 2a). This is because the time sweep was carried out at high temperatures (e.g. 160 °C), where the network relaxation is overwhelming at the measurement frequency (e.g. 2 rad/s).

During inverse vulcanization, the progressive crosslinking may result in the shrinkage of the sample in the parallel plate. Meanwhile, gas by-products such as $H_2S$ generated from the degradation of sulfur segments could lead to cavitation and inflation of the products[6]. Both the issues may disturb the measurement and cause modulus loss during the time sweep. To inspect the physical properties of the samples at different stages excluding potential effect of shrinkage and cavitation, four inverse vulcanization products were collected in vials at different reaction times (50, 100, 155, 240 min) according to the $G'$-$t$ curve (Fig. 2a), sequentially named Sample DIBS1-1, DIBS1-2, DIBS1-3 and DIBS1-4 (Table 1 and Supplementary Table 1), and hot-pressed into solid circular disk (thickness ~1.2 mm) for rheological measurement. The hot-pressing was carried out in 5 min to mitigate the error resulting from the evolution of inverse vulcanization products. Then the disk samples were subjected to strain sweeps to identify their LVE regimes at 10 rad/s and 110 °C, a temperature at which the dynamic exchange of S-S bonds is relatively slow. The strain sweep results (Supplementary Figs. 2–5) show that the sample at the curing stage and late over-cure stage has a broad linear regime beyond 50% strain with $G' > G''$, which indicates sparse crosslinks, while the samples at $G'_{max}$ and early over-cure stage show narrow linear region <1% strain, arguably due to the dense crosslinking.

Furthermore, to reveal the evolution of the polymer network and segment, frequency sweeps were performed for the four samples at temperatures from 110 to 160 °C at an interval of 10 °C (Supplementary Figs. 6–9). The $G'$ at 0.2 rad/s retrieved from frequency sweep curves at 160 °C was tabulated in Supplementary Table 2 and compared with those obtained from the $G'$-$t$ curve (Fig. 2a) at the sampling times. As can be seen, there is not much difference in the $G'$ values between the two tests, albeit $G'$ is slightly higher in the frequency sweep, but the

difference is significant for the sample in the late over-cure stage (240 min, DIBS1-4) where $G'$ in the frequency sweep is larger than that of time sweep by several orders of magnitude. More importantly, the frequency sweep test $G'$ experienced the same trend as the time sweep test $G'$, that is, rising first before declining with reaction time. Excluding the potential cavities through hot-pressing, the results confirmed that the change in $G'$ with reaction time is actually due to the polymer product evolution during the inverse vulcanization, although other factors such as inflation or shrinkage during the time sweep contribute to modulus loss in the late over-cure stage.

The frequency sweep data at different temperatures were superposed to establish $G$-$\omega$ master curves following the TTS principle taking 120 °C as the reference temperature ($T_r$, Fig. 2b). As shown, the superposition is successful over a broad frequency range, with the typical transition (i.e., $G'$-$G''$ crossover) from Maxwell-type liquids (i.e., $G' \propto \omega^2$ and $G'' \propto \omega$) to rubbery plateau (i.e., $G' \propto \omega^0$ and $G''$ having a shallow depth) with increased frequency clearly captured. Note that for DIBS1-1, at high frequencies, there seems some subtle failure regarding the superposition, which might be due to the polymerization of the residual alkenyl groups into crosslinks during frequency sweep at high temperatures.

Herein, the $G'$-$G''$ crossover is most likely the characteristic of dynamic covalent network relaxation realized by S-S bond exchange. This has been well recognized since the adaptable covalent networking microstructure has a strong correlation with temperature (frequency). The crossover of $G'$ and $G''$ is observed in $G$-$\omega$ curves of all samples (Supplementary Figs. 6–9) measured at 120 °C and higher temperatures, but does not exist in the curves at 110 °C. This indicates that, within the timescale of measurement, i.e., shorter than ~120 s (0.05 rad/s) which is comparable to the typical timescale of polymer processing, the dynamic exchange is regarded to be active at 110–120 °C. In low temperatures (110 °C) below the activation temperature of the adaptable covalent bonds (or a time shorter than its characteristic relaxation time, presumably corresponding to the $G'$-$G''$ crossover), the network microstructure is presented ($G' > G''$), while above the activation temperature (or a time longer than its characteristic relaxation time) the rheological response is suggestive of dissociated polymer ($G' < G''$).

The dynamic exchange mechanism of S-S can be inferred from vertical shift factors ($b_T$, Supplementary Fig. 10) in the superposition, i.e., the ratio between $G_N^0$ at the measurement temperature and that at the reference temperature[26,38]. In the frequency sweep results and $b_T$ plots (Supplementary Figs. 6–10), the plateau modulus increased (represented by $b_T$) as the measurement temperature was increased except for the DIB-S samples measured at 160 °C. The experimental $b_T$ are in consistency with theoretical $b_T$ dictated by entropy elasticity as Supplementary equation 1 expresses, which indicates the trace loss of crosslink density. The increased trend implies that the relaxation via S-S bond exchange majorly follows associate mechanism reported in

disulfide-crosslinked materials[39]. As the temperature was increased to 160 °C, the modulus contrarily declined, probably as a result of the loss of crosslinks, indicating the presence of sulfur radical species and dissociate exchange mechanism[26].

To discern the dynamics difference between the samples collected at different reaction stages, the $G'$-$\omega$ master curves are represented in terms of weighted relaxation spectra ($\tau^*H(\tau)$, Fig. 2c) via transformation according to a nonlinear regularization method proposed by Honerkamp et al. (Eq. 1)[40]

$$H(\tau) = G' \left[ \frac{d \log G'}{d \log \omega} - \frac{1}{2} \left( \frac{d \log G'}{d \log \omega} \right)^2 - \frac{1}{4.606} \frac{d^2 \log G'}{d(\log \omega)^2} \right]_{\tau = \sqrt{2}/\omega} \quad (1)$$

As shown in Fig. 2c, the weighted relaxation spectra are characterized by a prominent $\tau^*H(\tau)$ peak lying at timescales ranging from 6.23 s (DIBS1-1) to 24.92 s (DIBS1-2), which is ascribed to the network relaxation ($\tau_n$) via metathesis of S-S bonds. Generally, the characteristic relaxation time is an indicator of the relaxation behavior of some specified motion units (here the networks) and hence closely related to its structures and environments (e.g., plasticization effect of solvents), while the intensity of the relaxation peak is more indicative of the quantities of relaxation units.

The $\tau_n$ values of the four samples retrieved from the relaxation spectra are also listed in Table 1 to compare the adaptability of the network via S-S bonds exchange. The lowest $\tau_n$ and weakest peak intensity of DIBS1-1, collected in the early cure stage (50 min) before reaching the optimum cure stage, indicate its fastest relaxation process from the weakest network structure as well as its lowest network density. This is mostly due to the relatively high sulfur proportion and low crosslinks in the product since only a small amount of alkenyls are incorporated, resulting in a spare network with long sulfur chains between neighboring crosslinks. It is well-known that the higher the sulfur rank, the more prone the multi-sulfur moieties are to reversibly cleave and rebuild upon heating[22,41], i.e., faster relaxation.

As the inverse vulcanization advanced, the progressive reaction between sulfur radicals and alkenyls continuously consumed the unstable long sulfur chains into short sulfur chains, resulting in an increased crosslinking density with low sulfur rank. The crosslinking density was expected to reach its maximum (i.e., the sulfur chain between crosslinks is the shortest) when the networking became saturated, characterized by the moduli ($G'$ and $G''$) peak in Fig. 2a, namely the optimum cure stage. This was confirmed by the highest $\tau_n$ and the greatest $\tau H(\tau)$ intensity obtained in the sample of DIBS1-2, which was collected around the optimum cure stage (100 min). Even for the sample collected at 155 min (DIBS1-3) in the over-cure stages, the $\tau_n$ (18.67 s) was only observed to be slightly smaller than DIBS1-2 with a comparable relaxation intensity. However, for a long period of over-cure, as demonstrated in DIBS1-4 (240 min), the relaxation of the networks again became fast with a smaller $\tau_n$ (13.44 s) and an obviously lower relaxation intensity. This indicates a diminishment of crosslinking density and an easier network breaking in the over-cure stage, which is probably ascribed to the alternation of the sulfur chains between crosslinks and/or the plasticization by newly generated by-product molecules and oligomers.

As the rubbery plateau modulus ($G_N^0$) is commonly used to reveal the network structure, the $G_N^0$ of the inverse vulcanization products was determined from the master curves (Fig. 2b), as the $G'$ value at the frequency where $G''$ was at its minimum in the plateau zone[42], and listed in Table 1. Also listed is the molecular weight ($M_c$) between crosslinks calculated from the $G_N^0$ using Eq. 2[43]:

$$M_c = \frac{RT\rho}{G_N^0} \quad (2)$$

where $\rho$ is the density (g/cm³) of the sample, $R$ is the gas constant, and $T$ is the Kelvin temperature. $M_c$ is the constitutive parameter of a polymer network independent of the density and the temperature. The density was experimentally measured via the water flotation method and listed in Table 1. Since the dilation of the sample with increased temperature is negligible in light of the good linear relation between $\Delta T/T_r$ and $b_T$ in TTS (Supplementary Fig. 10 and affiliated discussion following the figure), the $M_c$ was calculated from $\rho$ and $G_N^0$ at $T_r$ (Table 1) ignoring the trace deviation of density. As shown, from DIBS1-1 to DIBS1-2, $G_N^0$ increases from 0.0446 MPa to 0.332 MPa, and the density rises remarkably from 1.272 g/cm³ to 1.324 g/cm³, further verifying the formation of compact and strong polymer networks after the curing stage. Meanwhile, the $M_c$ was significantly reduced from 93.2 kDa to 13.4 kDa, indicating the shortening of the sulfur chains between crosslinks in the networks. However, in the over-cure stage, both the density and the $G_N^0$ started to decline while the $M_c$ underwent a significant rise, see the data of DIBS1-3 and DIBS1-4 samples in Table 1. This confirms that the networks were loosed with diminished crosslinks formed upon continuous heating beyond the optimum cure stage. Note that the determined $M_c$ of the products here is comparable with the entanglement molecular weight ($M_e$) of many macromolecules[44], implying the possibility of chain entanglement, which may complicate the relaxation motions of the sample. However, since the typical hierarchical relaxation of entangled polydisperse branched polymer[45,46] is not observed in the frequency curves (Fig. 2b), the disentanglement behaviors are supposed to be covered by network relaxation via metathesis of the ubiquitous sulfur segments. This is further corroborated by the fact that at the test temperatures (120–140 °C), the disentanglement of common macromolecules such as polystyrene and poly(methyl methacrylate)[47] is too slow to be observed. Therefore, the effect of disentanglement is not considered in this case, which will be investigated in our future work.

The structure evolution of the polymer products with the reaction process is also characterized by the changes in gel fractions and $T_g$ (Table 1). The gel fraction increased from 60% to 90% when DIBS1-1 progressed to DIBS1-2, undoubtedly verifying the consumption of the alkenyl group into crosslinks fixing more polysulfur chains. Meanwhile, a rapid rise in $T_g$ (10 °C to 35 °C) in the curing stage validates the formation of a more compact and less flexible network. Likewise, albeit there is not much difference in the gel fraction and the $T_g$ between DIBS1-2 and DIBS1-3, they experienced an evident decline in the late over-cure stage between DIBS1-3 and DIBS1-4, from 87% to 75% and 39.0 to 36.0 °C, respectively. This again indicates the generation of small molecules and linear polymers from the degradation of inverse vulcanization products that induce plasticization effects. Moreover, by calculating network connectivity in terms of average coordination number ($z_{co}$) of polymer segments regarded as beads[48,49] (Supplementary Table 3 along with discussion), it is confirmed that the number of crosslink points increased followed by a decrease during inverse vulcanization, consistent with the change in $M_c$ (Table 1).

To further reveal if the alteration of the sulfur chain length upon heating dictates the change in relaxation and activation energy, samples (DIBS1-1, DIBS1-2, DIBS1-3, and DIBS1-4) were subjected to XPS investigation to have the chemical connections of the sulfur atoms unveiled (Supplementary Fig. 11). By deconvoluting the S 2$p$ signals, the length of sulfur chains between organic units, i.e., sulfur rank, is estimated from the ratio between signals of the S atom connected to carbon and those of the S atom connected to two sulfur[25] (Table 1). It is noted that the sulfur rank calculated from XPS (6–7) is significantly higher than the theoretical values. The high sulfur rank indicates that a lot of isopropenyl groups in DIB actually polymerized or transformed into terminated ends instead of connecting two sulfur chains, which also explained the large $M_c$ (>10 kDa) in all the samples (Table 1). In the view of the evolution of the sulfur chain, the sulfur rank experienced a decrease (6.7 to 5.96, DIBS1-1 to DIB1S2-2) during the curing stage due

to the progressive inverse vulcanization, which consumes the long sulfur chains. The following increase of sulfur rank back to about 6.6 in the over-cure stage, along with the appearance of another S 2$p$ signal with a binding energy of 164.7 eV suggests the transformation of the sulfur chain into another sulfur-rich moiety. Interestingly, element analysis (CHNS) showed (Supplementary Fig. 12) that the sulfur proportion in DIB-S declines (48.1% to 47.2%) slightly but continually during inverse vulcanization, which was usually argued to be ascribed to the generation of H$_2$S[6], in contrast to the alteration of sulfur ranks. The result confirmed that besides the release of H$_2$S there are other side reactions involving sulfur chains. The evolution in sulfur domains explains the alteration of relaxation behaviors that a drop in $\tau_n$ results from the increase in sulfur chain length as well as the generation of by-products as plasticizers.

On the other hand, the characteristic relaxation time $\tau_d$ of the polysulfide, estimated from the reciprocal of the frequency corresponding to the $G'$-$G''$ crossover in master curves (Fig. 2b), represents the timescale of the transition from rubbery elasticity ($G' > G''$) to liquid ($G' < G''$) via network relaxation mediated by S-S bonds metathesis. The S-S metathesis is a chemical reaction, and hence the temperature dependence of $\tau_d$ is expected to be described by the Arrhenius equation (Eq. 3):

$$\ln\tau_d = \frac{E_a}{RT} + \ln A \qquad (3)$$

where $E_a$ is the activation energy of the relaxation in kJ/mol, ln$A$ is the pre-exponential factor, $R$ is the gas constant, and $T$ is the Kelvin temperature. The ln$\tau_d$ of the four samples collected at different stages and tested at temperatures above 120 °C is plotted against the 1000/T in Fig. 2d. All the data except DIBS1-1 (50 min) are well fitted by the Arrhenius equation with $R^2$ ranging from 0.94 to 0.99 after the exclusion of the $\tau_d$ data at 160 °C (Fig. 2d) where dissociate mechanism makes impact on bond exchange (Supplementary Fig. 10 along with discussion). The relatively worse fitting of DIBS1-1 is probably attributed to the progressive inverse vulcanization of the unreacted alkenyl moieties at high test temperatures. Subsequently, $E_a$ and ln$A$ are determined from slope and intercept, respectively, and listed in Table 1. As shown, the $E_a$ increases from 108.9 to 133.7 kJ/mol as the inverse vulcanization proceeded from DIBS1-1 (curing stage) to DIBS1-2 (early over-cure stage). The significant increase of the activation energy or energy barriers for the network relaxation via S-S metathesis further confirms the structure evolution of long sulfur chains (DIBS1-1) into short sulfur chains (DIBS1-2,3) between crosslinks with strong S-S bonds that require more energy for metathesis. Under expectation, the $E_a$ of DIBS1-4 (late over-cure stage) was reduced to 131.3 kJ/mol with a significant drop in ln$A$ compared to the prior samples, indicating a smaller energy barrier for relaxation, which is due to the alteration of relaxation units such as length of sulfur chain or the generation of hanging chain ends and small molecules plasticizing the inverse vulcanization products[50,51]. In other words, as aforementioned, beyond the optimum cure stage, especially in the late over-cure stage, there can be evident alterations in relaxation units and product compositions, which make the network more spare and relaxed in an accelerated manner.

The inverse vulcanization process of DIB was also performed via time sweep at high temperatures (170 °C and 180 °C) and sulfur feed ratios (3 and 4) (Supplementary Figs. 13–16). All the inverse vulcanization products exhibit similar inverse vulcanization curves as the above model system (160 °C and sulfur feed ratio of 2). The higher temperature accelerates the reaction and structural evolution, resulting in negligible induction and shorter cure stages (about 10 and 30 min at 170 °C and 180 °C). Meanwhile, at high temperatures where the relaxations of polysulfides are activated markedly, moduli of the samples are relatively low and nearly unmeasurable in the late over-

cure stage. Likewise, the increased sulfur feed ratio also enhanced the reaction rates in all the stages, resulting in short induction, short curing, and early over-cure stages.

In summary, from time sweep and frequency sweeps, it is demonstrated that the inverse vulcanization of DIB can be divided into stages, including induction, curing, and over-cure beyond the optimum cure stage, which is accelerated at high temperature and regardless of the sulfur feed ratio. Throughout the stages, the polymer network is first developing towards a dense state with increased crosslinking density and shortening sulfur chains between crosslinks, while becomes weakened and loosed beyond the optimum cure stage, potentially due to the structure alternation (e.g., sulfur chains branching and/or generation of by-products). Correspondingly, the network relaxation experiences kinetics from fast to slow and then fast again, which is also characterized by an increase followed by a decrease in $T_g$ and gel fraction, and a converse alteration of sulfur rank (Table 1).

To verify if the observation from DIB is universal and the conclusion holds for other monomers, the inverse vulcanization of DVB, SO, and DCPD were also tested by time sweeps to draw their inverse vulcanization curves. DVB is an aromatic diene with vinyl groups instead of isopropenyl groups in DIB which tend to polymerize into bis-thiocumyl units[21]. Due to the less steric hindrance, DVB is likely to be more active in inverse vulcanization and homo-polymerization than DIB[52] to construct a polymer network with higher crosslinking densities. Because the vitrification of DVB in inverse vulcanization is too fast to remain homogeneous liquid state to be transferred, solid disk-shape samples were prepared for the rheological tests by hot-pressing (Fig. 1) the just vitrified DVB samples with a sulfur feed ratio of 2 (Supplementary Table 1). As shown in Supplementary Fig. 17, at 180 °C the inverse vulcanization product of DVB (Sample DVBS2) experienced a similar evolution pattern as DIB with a short cure stage followed by an over-cure stage, having a liquid-like product ($G' < G''$) finally obtained (Supplementary Fig. 17) as a result of crosslinks loss. Intriguingly, when inversely vulcanized at 160 °C the time sweep of DVB (Sample DVBS1, Supplementary Fig. 18) was featured with a monotonical increase of $G'$ before reaching a plateau (optimum cure stage) over the rest of measurement time without decline, suggesting negligible degradation of the network in the over-cure stage. The difference in inverse vulcanization curves between DVB and DIB implies the existence of a temperature threshold, above which the degradation of polysulfide networks becomes significant as a result of moduli loss in the over-cure stage. The temperature threshold is expected to depend on the structure of organic comonomers.

SO, a triglyceride, was also used as a typical olefin-like comonomer with multiple alkenyl moieties to investigate the inverse vulcanization. The alkenyl groups in SO are always di-substituted by alkyl chains at two different carbons without π-π conjugation, in great contrast to aromatic crosslinkers such as DIB and DVB. The inverse vulcanizations of SO were performed at varied temperatures (160, 170, and 180 °C) with higher sulfur feed ratios (4 and 6) due to the relatively sparse alkenyl bonds (Supplementary Table 1, Fig. 3a and Supplementary Figs. 19–22). At 160 °C, the inverse vulcanization curve of SO includes typical induction and cure stages followed by a long plateau over-cure stage regardless of the sulfur feed ratio (Sample SOS1 and SOS2, Supplementary Figs. 19, 20), and only a minor drop in $G'$ is observed near the end of the test (~230–260 min). As the reaction temperature is 170 or 180 °C (SOS3, SOS4, and SOS5), a drop of moduli is observed at a long reaction time in the over-cure stage (Fig. 3a and Supplementary Figs. 21 and 22). Interestingly, a slight increase in $G'$ appears in the late over-cure stage of SOS4 and SOS5. The over-cure of poly(SO-S) leads to a decline in $G'$ by two orders, which is significantly less than that (six orders) of DIB inverse vulcanization while slightly less than that of DVB, indicating a different degree of degradation.

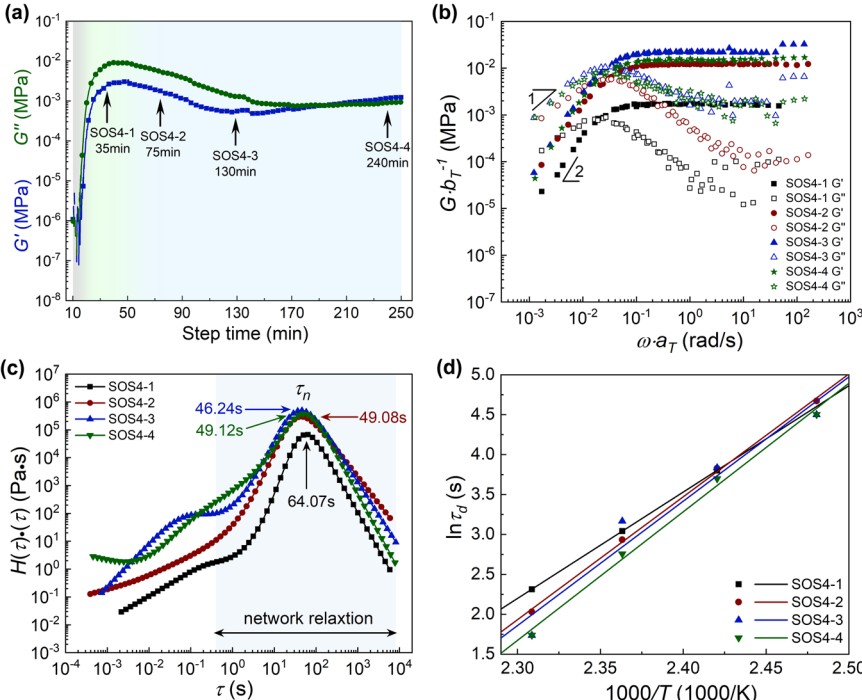

**Fig. 3 | Rheological test results of SO inverse vulcanization. a** The time sweep test of the SOS4 sample at 1 rad/s at 180 °C. **b** The master curves of SOS4 samples at different reaction times. **c** Weighted relaxation time spectra of the investigated samples. **d** Fitting of the characteristic relaxation time ($\tau_d$) of SOS4 samples at different reaction times to an Arrhenius-type equation.

To exclude any potential effect of sample cavitation and change in density at 180 °C and to elucidate the abnormal increase in $G'$ (Fig. 3a), a solid disk specimen was used for the frequency sweeps based on the samples collected at a reaction time of 35, 75, 130, and 240 min (Supplementary Figs. 23–26), named SOS4-1, SOS4-2, SOS4-3, and SOS4-4 respectively. Following the same analysis procedure as the inverse vulcanization of DIB, $G$-$\omega$ master curves are constructed based on the frequency sweep data (Fig. 3b and Supplementary Fig. 27), and the weighted relaxation spectra are thereby plotted (Fig. 3c). The activation temperature of dynamic exchange reaction mediated by polysulfur bond is estimated to be 120–130 °C on basis of the appearance of $G'$-$G''$ crossover in the $G$-$\omega$ curves measured at 130 °C rather than 120 °C (Supplementary Figs. 23–26). The monotonous increase of $b_T$ with temperatures (Supplementary Fig. 27) demonstrates the associate mechanism[39] is dominant in the relaxation of dynamic SOS via S-S bonds.

As shown in Fig. 3b, all the products at all the stages exhibit a typical rubbery plateau, while surprisingly, they can dissolve completely in tetrahydrofuran (THF) (Table 1), indicating the plateau is a characteristic of entanglement networks instead of chemically crosslinked networks. Nevertheless, their rubbery plateau regions are plain and wide, completely different from the entangled polydisperse branched polymers, which usually have hierarchical relaxations[45,46] and a sloping rubbery plateau[53]. Since the disengagement time, $\tau_d$ of polymer is linearly dependent on $N^3$ or $Z^3$[38], where $N$ and $Z$ are the polymerization degree and the number of entanglements per molecule respectively, we argue that the entanglement relaxation of poly(SO-S), branched polymers with high molecular weight generated from the multi-alkenyl monomer as the branch point, is too slow to be observed and covered by the relaxation mediated by S-S metathesis. Thereupon, in a poly(SO-S) system, both the entanglement and branched points serve together as knots in the mesh to construct the network structure of the inverse vulcanization products, and the entanglement that cannot relax in the test timescale can be regarded as pseudo-permanent crosslinks. The molecular weight $M_c$ calculated

from Eq. (2) is expected to be the average molecular weight of elastic chains between entanglement and branched points, which contributed to the vulcanized rubber-like behaviors in SAOS (Fig. 3b). In addition, the poor superposition due to the unexpected increase in modulus at high frequency (>$10^1$ rad/s) is likely to result from nonaffine elasticity and nonaffine displacements of polymer segments as revealed in the reference[54].

The relaxation time $\tau_n$ of the entanglement network mediated by S-S metathesis, $T_g$, and plateau modulus $G_N^0$ along with the calculated $M_c$ are listed in Table 1. As the inverse vulcanization proceeded, from SOS4-1 (reaction time 35 min, curing stage) to SOS4-2 (75 min, early over-cure stage), both around the optimum cure stages, an increase in relaxation intensity of $\tau_n$ along with rising $G_N^0$ and $T_g$ is observed, indicating a progressive network densification in the curing stage similar with poly(DIB-S). However, different from the increased density of poly(DIB-S), the significant increase in $G_N^0$ (0.0017 MPa to 0.012 MPa) of the inversely vulcanized SOS4 is accompanied by a slight decrease in density (1.067 g/cm³ to 1.058 g/cm³), meaning that instead of chemical crosslinking the structural evolution of the sample during the curing stage is more likely due to the progressive branching. The branching also results in an appreciable decline of $M_c$ from 237 kDa to 33.3 kDa, indicating the generation of more knots in the entangled branched polysulfides from the view of sulfide metathesis. Meanwhile, the $\tau_n$ shifts to a shorter timescale (64.07 s to 49.08 s). Subsequently, in over-cure stages from SOS4-2 (75 min) to SOS4-4 (240 min), $T_g$, $G_N^0$, $\tau_n$, and relaxation intensity did not change significantly (Table 1) with only a small fluctuation of $G_N^0$, indicating a slow and minor degradation of the sample. Moreover, different from DIB-S samples, in the late over-cure stage, the density of SOS remains increased (1.058 to 1.106 g/cm³) despite the fluctuation of $G_N^0$. This suggests the formation of a compact and rigid structure in late over-cure stages, consistent with the slight $G'$ increase observed in the time sweep (Fig. 3a).

The rearrangement of the network is further evaluated by the characteristic time $\tau_d$ retrieved from frequency sweep curves (Fig. 3b) and fitted by the Arrhenius equation (Eq. 3, Fig. 3d). An increase in $E_a$

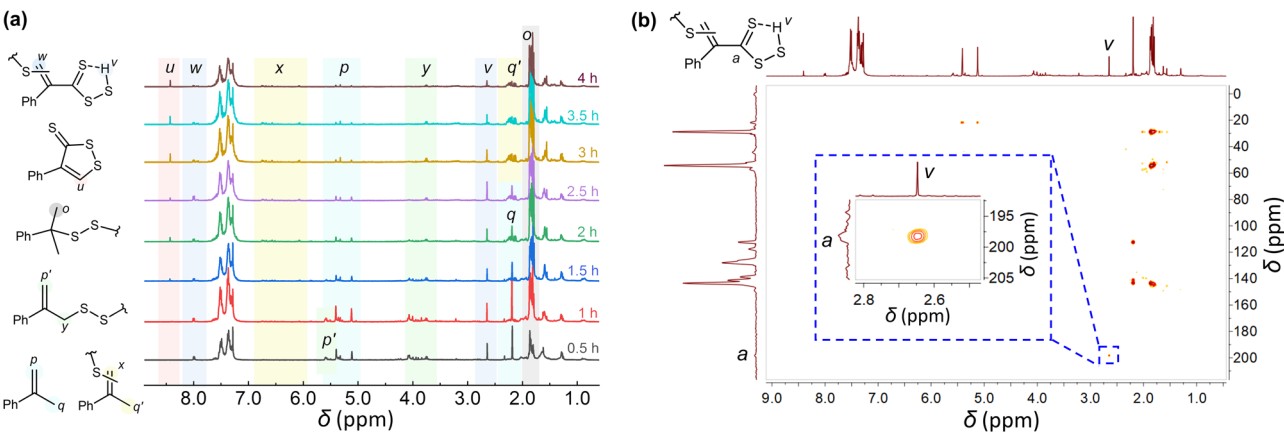

**Fig. 4 | NMR spectra of αMSt inverse vulcanization products. a** [1]H NMR spectra of αMSt samples at 160 °C with a sulfur feed ratio of 2 and different reaction times. (Solvent: CDCl₃). **b** HMBC spectrum of αMSt samples reacted for 1 h at 160 °C with a sulfur feed ratio of 2. (Solvent: CDCl₃).

(Table 1) from 110.7 kJ/mol to 127.9 kJ/mol is observed upon transition from the curing stage over the optimum stage. The remarked increase of $E_a$ verified the formation of short sulfur chains in cure stages, which coincides with the inverse vulcanization of DIB. The inference is also confirmed by XPS investigation (Supplementary Fig. 28), where a severe drop in sulfur rank is observed (7.24 to 6). Subsequently, an increase of $E_a$ is observed in the over-cure stage, corresponding to the slow degradation exhibited by retarded evolution of $\tau_n$ in weighted relaxation spectra (Fig. 3c) and time sweep but fluctuation in modulus in the late over-cure stage (SOS4-4) (Fig. 3a), which suggests a different evolution pattern in SOS. The sulfur proportion determined by element analysis (Supplementary Fig. 12) experiences a slight drop (38.0% to 36.9%) during inverse vulcanization similar to DIB, suggesting the limited generation of H₂S.

In sum, compared to poly(DIB-S), the structural evolution of poly(SO-S) especially during the over-cure stage is much slower and minor, and the change in density and $E_a$ is different, which needs an explanation. The different and complex inverse vulcanization curve of SOS, on behalf of aliphatic olefins-like monomers, is most likely due to the uncrosslinked nature, which also makes the poly(SO-S) evolve in a different pattern.

Lastly, DCPD, a specific monomer with cyclic alkenyl moieties, is inversely vulcanized at varied temperatures (160, 170, and 180 °C) and sulfur feed ratios (2 and 4) (Supplementary Table 1 and Supplementary Figs. 29–32). It is surprising that regardless of the temperature and the feed ratio, poly(S-DCPD) exhibits no loss of $G'$ after reaching the optimum cure stage in its inverse vulcanization curves (time sweep). Meanwhile, nearly no poly(DCPD-S) dissolved in THF after 24 h, suggesting the absence of linear or branched polymer fraction in the product, coinciding with rheological measurement on verifying the negligible degradation. The trace degradation is also confirmed by XPS spectra and element analysis (Supplementary Figs. 12, 33) that decreases in sulfur rank, sulfur proportion, and hydrogen proportion are observed without the appearance of distinct sulfur moieties, indicating that the generation of H₂S is supposed to the only side reaction during inverse vulcanization of DCPD. That is, the inverse vulcanization product of DCPD has superior stability against degradation under thermal treatment, offering a long period of optimum cure stage.

### Evolution of inverse vulcanization products and their kinetics

Regarding the different evolutions of the various inverse vulcanization products, as demonstrated in the last section, here comes a question: what are the reactions leading to the evolution of the polysulfide products, especially the degradation in the over-cure stage at high temperatures? More importantly, the different inverse vulcanization patterns among DIB, DVB, SO, and DCPD urge a reasonable mechanism

to elucidate the evolution variety depending on alkenyl structures. To catch a glimpse into the reaction mechanism, mono-function alkenyls are exploited as model monomers to prepare oligomers available for solution NMR analysis. The monomers in mechanism studies are expected to have a boiling temperature at least higher than 160 °C to reduce the error caused by monomer evaporation at reaction temperatures. α-methylstyrene (αMSt) and 4-methylstyrene (4-MSt) (Supplementary Table 4) are mono-function alkenes bearing aromatically conjugated isopropenyl and vinyl groups, respectively, resembling DIB and DVB. Allyl phenyl ether (APE) (Supplementary Table 4) represents monomers with isolated mono-substituted alkene groups. Although having two alkenyl moieties, ethyl linoleate (EOD) and sorbic acid (SA) (Supplementary Table 4) are chosen as model monomers for unsaturated aliphatic ester due to the good solubility of their inverse vulcanization products in solvents and a relatively high content of alkenyl bonds easier for analysis.

The reaction between elemental sulfur and αMSt is first investigated at 160 °C with a sulfur feed ratio of 2. After the reaction began, a batch (0.2 g) of sample was separated from the reaction mixtures for every half hour and subjected to NMR analysis. The [1]H NMR of samples are stacked in the sequence by reaction time (Fig. 4a). In the spectra, the sharp signals with chemical shifts of 5.0–5.5 ppm and 2.2 ppm belong to the isopropenyl group (H[p] and H[q]) in the αMSt monomer. Meanwhile, strong signals (H[o]) at 1.8 ppm are ascribed to the methyl group of the cumyl unit in oligomers[21]. The proton with a chemical shift of 8.45 ppm (H[u]) was demonstrated to be the proton signal of 4-phenyl-1,2-dithiole-3-thione (4-phenyl DTT), which was separated by column chromatography and identified by [1]H NMR, [13]C NMR and HMBC investigation (Supplementary Figs. 34–36)[55]. Besides the monomer signals and main product signals reported, a noticeable signal (H[v]) at 2.6 ppm is observed throughout all the spectra. To reveal the identities of the suspicious signal, the 1 h sample is subjected to HMBC experiments to detect heteronuclear long-range correlation, and the resultant spectrum of the sample is present in Fig. 4b. It is found that the H[v] is strongly correlated with a carbon signal at 198 ppm (C[a]). The high chemical shift suggests that the carbon is supposed to exist in a carbonyl-like group. Considering that the experiments were carried out in a closed environment with minor exposure to oxygen contrary to a noticeable signal of C[a] as well as its neighboring group (H[v]), C[a] is identified as the carbon in thiocarbonyl moieties (Fig. 4b). This ascription is supported by chemical shifts of thiocarbonyl groups present in various molecules, which range from 190 ppm to 260 ppm[56,57]. The existence of the thiocarbonyl group is also corroborated by the similar thiocarbonyl domain in DTT, whose chemical shift is higher (~220 ppm), probably due to the cyclization into the dithiol ester, which could be aromatic. To further identify the proton

signal $H^v$ (Fig. 4b), HSQC experiments are exploited to detect the carbon connecting directly to proton $H^v$, but it is found that there is no correlation between the $H^v$ signal and any carbon signals (Supplementary Fig. 37). The result denotes that the $H^v$ is probably a thiol proton instead of a C-H proton. Considering that in dithiocarboxylic acids, the thiol group connected to the thiocarbonyl group always has a chemical shift larger than 6 ppm[58], the $H^v$ should be the proton in a carbo(dithioperoxo)thioic acid (CDTA) group (Fig. 4b). The presence of the thiocarbonyl group in CDTA also rationalizes the appearance of doublet signals at 7.9–8.0 ppm ($H^w$, Fig. 4a), which diminished simultaneously with $H^v$. $H^w$ is believed to be protons in the alkenyl group conjugated with thiocarbonyl groups, resulting in the shift of signal to the low field compared to common alkenyl protons, which will be confirmed in inverse vulcanization product of 4-MSt. Moreover, the CDTA could be an intermediate product from the degradation of sulfides into DTT, which will be elucidated in the mechanism part. Lastly, a broad signal band ($H^y$) around 3.3–4.2 ppm appeared together with $H^{p'}$ at about 5.3–5.5 ppm in the spectra of 0.5 and 1 h samples but diminished simultaneously in the samples with longer reaction time. In the mechanism reported by J. Pyun[21], there are intermediates containing isopropenyl moieties substituted by a sulfur chain at the methylene group, which can account for the $H^y$ and $H^{p'}$ signals (Fig. 4a)[59]. Hence both signals diminished as the consumption of alkenyl groups continued. Similarly, sporadic signals at 6.0–6.8 ppm ($H^x$) and 2.2 ppm ($H^q$) are argued to be ascribed to other sulfur-substituted abstraction products (Fig. 4a).

So far so good, all the signals influenced by sulfur substitution at the low field in $^1$H NMR spectra have been identified, and thiocarbonyl moieties are demonstrated to exist in CDTA form beyond DTT, which may affect the properties of inverse vulcanization products. To track the evolution of products, the proton signals, $H^p$, $H^q$, $H^u$, and $H^v$, are integrated and normalized to phenyl signals (excluding CH₃Cl signals) as the reference. Thereafter, the amount of remaining αMSt (1-conv$_{MSt}$) can be calculated from the integral of $H^p$, while the integral of $H^o$ estimates the amount of cumyl group ($Y_{cumyl}$), which is supposed to diminish due to the progressive side reaction such as H-abstraction or thiol elimination. On the other hand, the integrals of $H^u$ and $H^v$ represent the yield of units containing DTT and CDTA domains ($Y_{DTT}$ and $Y_{CDTA}$) regarding αMSt units, respectively. The yields and residue amounts are plotted against time accordingly (Fig. 5). Similar to inverse vulcanization curves obtained by the rheology method, there are stages observed in the kinetic plot (Fig. 5). According to $^1$H NMR spectra, the αMSt monomer (up-pointing triangle, blue) is consumed in about 1.5 h at 160 °C with a sulfur feed ratio of 2 (Fig. 5a), which could be designated as the curing stage including a non-detectable induction period. Meanwhile, the ratio of cumyl signal reached its maximum as αMSt conversed. After a short plateau where the composition of the system remains almost unchanged, the amount of cumyl groups begins to drop, indicating a progressive H-abstraction in the over-reaction stage analogy to over-cure. It is noted that the identification of stages depending on conversions of monomers is not strictly the same as those identified by time sweep, since the reaction of monomer at low conversion would not enhance the modulus of the sample, while a full conversion of monomers does not strictly match the max $G'$ achieved in time sweep. In the view of by-products, DTT began to be captured when αMSt reached high conversion (1 h), and then gradually increased in the rest time. What is surprising is that there is a large amount of CDTA units (8%) in the product in the curing stage (0.5 h) earlier than the appearance of DTT, and its yield significantly decreased as the reaction continued while the DTT began to accumulate. The consumption of CDTA ($H^v$) together with the continuous generation of DTT ($H^u$) suggests that the CDTA could be the intermediate for the formation of DTT.

By changing the sulfur feed ratio and reaction temperature, we further inspect their effects on the inverse vulcanization kinetics. It is found that when the sulfur feed ratio increased from 2 to 4 (Fig. 5a–c), the inverse vulcanization consumed αMSt in a faster manner, shortening the curing stage from about 2 to 1 h. This is because the enhanced sulfur feed ratio actually increases the sulfur concentrations but decreases the concentration of αMSt. Therefore, the high concentrations of sulfur radical accelerated the consumption of the αMSt. Moreover, it is interesting that when the sulfur feed ratio increased to 3 or 4, the generation of DTT products is promoted with higher yields (Fig. 5a–c, orange squares at bottom) as the CDTA transformed to DTT faster (green down-pointing triangles), leaving a smaller CDTA concentration peak in the induction periods. It is noted that the yield of DTT is >10%, referring to αMSt units, indicating a large amount of by-product units. In the meantime, the H-abstraction altering the cumyl groups is comparable in three inverse vulcanizations. These results show that an increased sulfur feed ratio may accelerate the degradation of sulfide units, which suggests the formation of DTT involving excess sulfur.

On the other hand, by changing the inverse vulcanization temperature, the kinetics of inverse vulcanization also alter. As the temperature increased from 140 °C to 150 °C and 160 °C (Fig. 5d–f), the inverse vulcanization accelerated with a reduced induction period (4.5 to 2 h). As for side reactions, it seems that the lower temperature significantly suppressed the generation of DTT, with the $Y_{DTT}$ at 7 h declining from about 12% to 2% (Fig. 5d). However, a large amount of CDTA can be captured in the curing stage at low reaction temperature (140 °C), which means that the CDTA is more stable at the temperature and retarded to transform into other species including DTT. Concerning H-abstractions, although the effect of temperature is more complicated over a long time, it is clear that the H-abstraction, which already began to degrade the cumyl units in the curing stage, cannot be eliminated by simply lowering the temperature (Fig. 5d). Hence a lower temperature inversely results in a smaller $Y_{cumyl}$, i.e., more H-abstraction, at the end of the main cure stage compared to 150 °C and 160 °C systems. Nevertheless, the results show that reaction rates of the side reactions, including H-abstraction and the generation of DTT and CDTA, heavily depend on the temperature, which brings about the different inverse vulcanization curves obtained by the in situ time sweep of inverse vulcanization systems at varied temperatures.

Although the reaction rate of αMSt is distinct from DIB, owning to different viscosity and concentrations of reactants, the kinetic plot reveals that the development of organic units plays an important role in the evolution of inverse vulcanization products. The soluble fraction of poly(DIB-S) and poly(SO-S) was also subjected to NMR characterization. In the $^1$H NMR spectrum of the soluble fraction of DIBS1-4 (Supplementary Fig. 38), a signal -8.4–8.5 nm ($H^u$) validates the existence of DTT moieties in polymers, which is also confirmed by its XPS spectrum (Supplementary Fig. 11d). Meanwhile, the strong correlation between the proton signal at 2.60 ppm and the carbon signal at 198 ppm in the HMBC spectrum confirmed the existence of CDTA ($C^a$ and $H^v$) moieties in poly(DIB-S) (Supplementary Fig. 38). Hence, the formation and accumulation of by-products containing DTT moieties in polymers via CDTA probably account for the degradation of the network because of its good stability arising from cyclic structure. Moreover, a prevalence of thiocarbonyl moieties (CDTA and DTT) in inverse vulcanization products of αMSt as well as the progressive H-abstraction resulting in the loss of cumyl group and generation of sulfur-substituted isopropenyl groups may not only lead to degradation of polymer but also change the segment structures, therefore inducing the change of relaxation behaviors.

To confirm the prevalence of the degradation patterns in inverse vulcanization, we characterized products generated from elemental sulfur and various alkenes, including 4-MSt, APE, and EOD by NMR. In $^1$H NMR and HMBC spectra (Supplementary Figs. 39, 40) of 4-MSt reacted with elemental sulfur for 24 h, the strong proton signal at 2.6 ppm correlated with a carbon signal at about 198 ppm ($C^a$) verified the existence of CDTA-like moieties conjugated with an aromatic

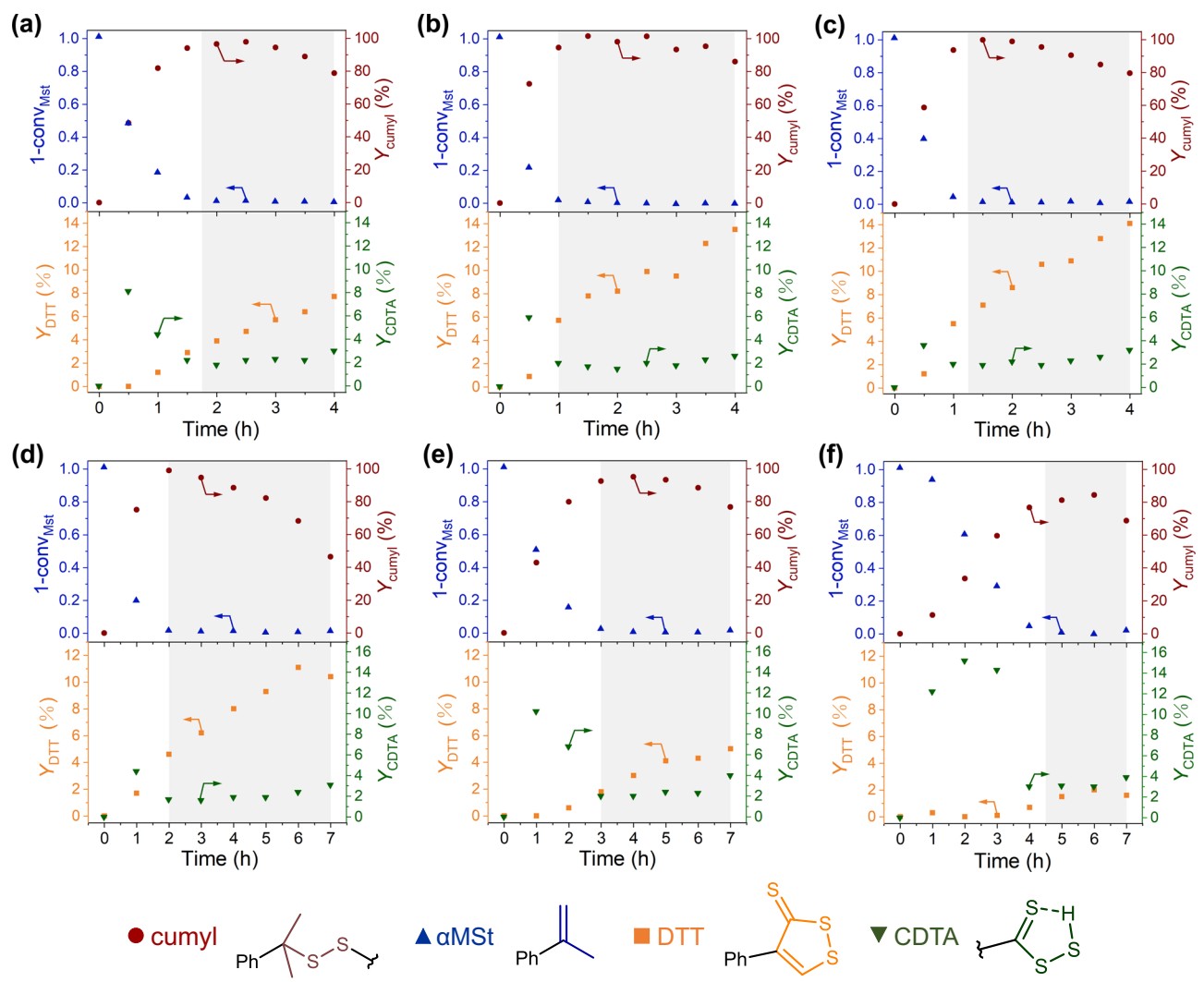

**Fig. 5 | Kinetic plots of αMSt inverse vulcanization under different reaction conditions.** The subfigures at the top display the inverse vulcanization with varied sulfur feed ratios of **a** 2, **b** 3, and **c** 4 at the same reaction temperature (160 °C). The subfigures at the bottom display the inverse vulcanization at varied reaction temperatures of **d** 140 °C, **e** 150 °C, and **f** 160 °C with the same sulfur feed ratio (2) and longer time (7 h). The 1-conv$_{Mst}$ represents the residue amounts of αMSt monomer estimated from H$^p$; $Y_{cumyl}$ represents the yield of cumyl group calculated from H$^o$; $Y_{DTT}$ represents the yield of DTT units (H$^u$); and $Y_{CDTA}$ represents the yield of CDTA (H$^v$). All the yields refer to αMSt units.

group. The thiocarbonyl carbon (C$^a$) signals are also correlated with hydrogen signals at 7.9 ppm (H$^w$), which was supposed to be proton at the alkene conjugated with thiocarbonyl groups, therefore rationalize the ascription raised above with regard to Fig. 4a. Meanwhile, proton signals at 8.2 ppm in the spectra of APE reacted with S$_8$ (Supplementary Fig. 41) corroborated the formation of alkenyl groups conjugated with thiocarbonyls similar to DTT and CDTA. The results point out that the formation of thiocarbonyl species is a remarkable side reaction in the inverse vulcanization of monomers with vinyl, isopropenyl, and propenyl groups, especially when they are conjugated with acryl rings.

On the other hand, the distinct proton signals at 6.56 and 6.89 ppm in the spectra of the soluble fraction of inversely vulcanized EOD reveals a different form of by-product structures in inverse vulcanization products (Supplementary Fig. 42). In the HMBC spectrum of the poly(EOD-S) product (Supplementary Fig. 43), the proton signals with a chemical shift of 6.56 ppm (H$^n$), whose chemical shift is higher than typical alkenyl protons in olefins (about 5.4 ppm)[17], is correlated with an aryl carbon (C$^e$). Because the aryl domain is absent in EOD feedstocks, the H$^n$ probably belongs to the proton at thiophene moieties[60], which also explains the correlation between methylene proton signal H$^m$ and acryl carbon (C$^e$ and C$^f$). Meanwhile, the H$^l$ signals

(Supplementary Fig. 43) and weak signal around 6.8 ppm (Supplementary Fig. 42) are suggested to be ascribed to the proton at or near the sulfur-substituted thiophene moiety. Similarly, in the $^1$H NMR spectrum of the poly(S-SA) (Supplementary Fig. 44), the thiophene signals are observed in large quantities. In the spectrum of poly(SO-S), there are weak signals at 8.1 ppm owing to the generation of a few DTT moieties (Supplementary Fig. 45). However, the strong signal at 6.57 ppm and a series of weak signals at 6.6–7.1 ppm indicate the existence of thiophene groups, which is confirmed by the correlated carbon signals (C$^e$) in the HMBC spectrum (Supplementary Fig. 46). The generation of thiophene moieties is also supported by the small peaks at 1436 cm$^{-1}$ and 964 cm$^{-1}$ appearing in the IR spectrum of SOS4 (Supplementary Fig. 47)[61]. The appearance of the thiophene implied that the degradation of inverse vulcanization product of triglycerides as well as olefins containing alkenyl groups in the middle of the chains follows the different mechanism into thiophene by-products in contrast to phenyl-conjugated alkene monomers.

### Mechanism and DFT studies
The above discussion illustrated the role of thiocarbonyls and thiophenes in the evolution of inverse vulcanization products, but the

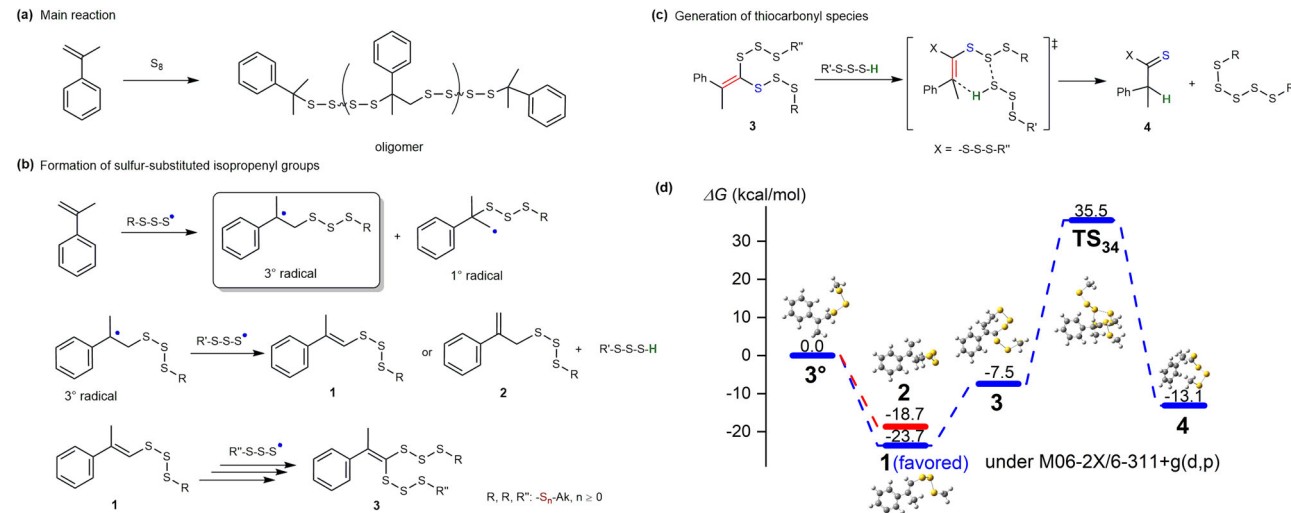

**Fig. 6 | Reactions in inverse vulcanization of αMSt. a** The major product of inverse vulcanization of αMSt; **b** Generation of sulfur-substituted alkenes via radical intermediates. **c** Rearrangement of sulfur-substituted alkenes into thiocarbonyl species in two proposed routes. **d** The whole profiles of Gibbs free energy barriers in the evolution of radicals into sulfur-substituted alkenes and then thiocarbonyl species in inverse vulcanization of αMSt. The temperatures applied for the scheme are around 140–180 °C.

mechanism of how thiocarbonyls and thiophenes species generated from organic sulfide remains elusive. Nevertheless, in the kinetic plot of αMSt inverse vulcanization (Fig. 4a), the presence of sulfur-substituted isopropenyl groups in two forms is revealed, which is also investigated using DFT techniques by Pyun et al.[21]. The sulfur-substituted alkenes in various forms could be the essential intermediates for the formation of thiocarbonyl and thiophenes via tautomerism-like rearrangement and cyclization, respectively, which provide a diagram for the evolution of inverse vulcanization products. The reaction routes in inverse vulcanization of monomer with alkenes conjugated with acryl groups and alkenes in olefins are illustrated in Figs. 6–8 as follows. Herein the inverse vulcanization of αMSt (Fig. 6a) is presented as an example.

Firstly, the reaction between sulfur radical and alkenes followed by H-abstraction generated an alkene domain substituted by sulfur chain (Fig. 6b). The addition by sulfur radical towards isopropenyl groups results in primary radical and tertiary carbon radicals. In the presence of another sulfur radical, the carbon radicals experience a H-abstraction reaction into sulfur-substituted alkenes species, in which the sulfur chains substitute at the alkene carbon (**1**) or the α-carbon of the alkenes (**2**)[21]. The isopropenyl signal of species **2** is distinguishable in the $^1$H NMR spectra in Fig. 4a as $H^{p'}$ and $H^y$, and species **2** will be further consumed by sulfur radical in the inverse vulcanization (Fig. 4a). In contrast, species **1** is expected to be more favored to be generated due to the direct conjugation of sulfur with alkene carbon, therefore enabling its identification by $H^x$ in the NMR spectrum throughout the kinetics studies (Fig. 4a). With repeat radical addition followed by H-abstraction, species **1** becomes a sulfur-disubstituted alkenes **3**. The sulfur-disubstituted alkenes **3**, which is unstable to be captured in experiments, is the essential intermediate for the generation of thiocarbonyls. In the presence of sulfhydryl generated from H-abstraction by sulfur radical, the active species **3** can form a six-membered ring transition state (TS) with the sulfhydryl thereby, the sulfur-substitute alkene rearranges into a thiocarbonyl species **4** (Fig. 6c). When the thiocarbonyl is connected with a sulfur chain, a dithioester moiety is obtained, which is the precursor of CDTA. It is obvious that the route involves a large amount of sulfur atoms in both substituted alkenes and sulfhydryl, which explains the fast degradation rate at high sulfur feed ratios observed in Fig. 5a–c. On the other hand, the high sulfur rank in inverse vulcanization product of DIB and αMSt due to the

transformation of isopropenyl groups into mono-substituted isopropyl domain favors the reaction, leading to a remarked degradation.

The DFT calculations were employed to investigate the process (Fig. 6d). Formation of species **1** and **2** from original radical species was confirmed as a spontaneous process since lower Gibbs free energies of products were obtained (−18.7 and −23.7 kcal/mol, radical species was set as 0), where species **1** was preferred to produce due to better stability compared with species **2** ($\Delta G$ = 5.0 kcal/mol). The relatively low energy of both species **1** and **2** allow their identification in NMR spectra as unstable intermediates (Fig. 4a). After a radical addition process, the generated species **3** cyclized via a six-membered ring transition state (**TS$_{34}$**) with $\Delta G_{34}$ = 43.0 kcal/mol, into thiocarbonyl species **4** with relatively good stability with falling energy (−13.1 kcal/mol) compared to starting radicals.

As the active thiocarbonyl species **4** is generated, CDTA and DTT will derive from it subsequently (Fig. 7). Due to the neighboring dithioester group and phenyl group, their tertiary α-carbon is more active for H-abstraction into an alkenyl moiety conjugated with both thiocarbonyl and phenyl group (species **5**, Fig. 7). After another sulfur radical addition followed by H-abstraction to recover the alkenyl group, the species **5** evolves into species **6** (Reaction series *i*). Lastly, a hydrogen transfer reaction from free sulfhydryl species breaks the sulfur chain connected to the thiocarbonyl group, yielding CDTA (species **7**, Reaction series *ii*), whereby the sulfhydryl proton forms an intramolecular five-membered ring configuration thanks to hydrogen bonds. It should be pointed out that reaction *ii* is independent of reaction series *i* and could happen at any stages as long as the thioester group forms. Hence, the CDTA is stabilized and the species bearing CDTA moieties are able to be identified in the products with strong $H^v$ signals, as Fig. 4 shows. Nevertheless, the CDTA moieties can further cyclize into the most stable DTT (species **8**, Fig. 7b) via a simple four-membered ring TS. Finally, the by-products bearing the DTT ring are obtained and identified in inverse vulcanization products of αMSt, 4-MSt, DIB, etc., as long as the monomer consists of an alkenyl group activated by phenyl groups. It is very interesting that the DTT is expected to be an aromatic group, with every sulfur atom contributing a paired electron from its 3*p* orbital, according to the DFT calculation result. The aromatic nature explains the stability of DTT as a major by-product in inverse vulcanization. In addition, the excessive sulfhydryl ends generated from repeated H-abstraction can combine into a

**Fig. 7 | Reactions bringing about CDTA and DTT. a** Repeated H-abstraction and hydrogen transfer of the intermediate containing dithioester group into CDTA; **b** Cyclization of CDTA into DTT; **c** Release of H₂S from sulfhydryl ends as H-abstraction products. **d** The whole profiles of Gibbs free energy barriers in the evolution of dithioester intermediates into CDTA and DTT by-products. The temperature applied for the scheme is around 140–180 °C.

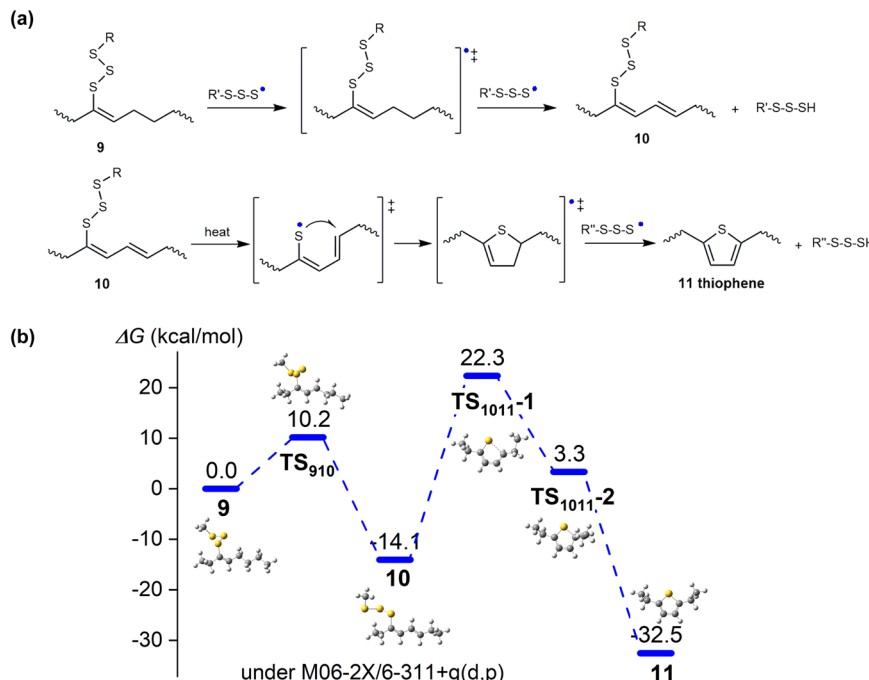

**Fig. 8 | Reactions bringing about thiophenes. a** Evolution of sulfur-substituted alkenes into thiophene. **b** The whole profiles of Gibbs free energy barriers in the evolution of sulfur-substituted alkenes into thiophene via cyclization in inverse vulcanization of olefin-like monomers. The temperature applied for the scheme is around 160–180 °C.

longer sulfur chain with $H_2S$ released, explaining the origin of $H_2S$ by-products.

The derivation reactions starting from species **4** were also investigated by DFT calculations (Fig. 7b). The proton radical extraction and proton extraction process from species **4** to species **6** was swift, with only a low Gibbs free energy barrier occurring at first proton extraction (**TS₄₅**) of 10.4 kcal/mol, with energy falling since conjugation expansion into $\alpha$, $\beta$-unsaturated thioketone. After the mercaptan exchange, CDTA was obtained with slightly lower energy ($\Delta G = -2.3$ kcal/mol), having a relatively weak hydrogen bond formed between the mercaptan end and thioketone. DTT was generated from CDTA by a transition state **TS₇₈** of intramolecular cyclization between mercaptan groups with $\Delta G_{78} = 33.9$ kcal/mol, as the rate-determining step in this stage. The relatively slow transformation of CDTA into DTT but also the low energy of CDTA compared to other intermediates (species **1**–**6**) enable the clear identification of CDTA domains by NMR as discussed above. It is worth noticing that DTT was ensured with strong stability (−41.9 kcal/mol) due to the aromatic heterocycle.

Contrary to inverse vulcanization of aromatically conjugated alkenes, inverse vulcanizations of monomers containing isolated alkenyl groups such as olefins and triglycerides proceed in distinct manners, which have been revealed by rheological and NMR investigations above. The thiophene has been demonstrated as the major degradation product in inverse vulcanization of fatty acids and triglycerides. The generation of thiophene from similar sulfur-substituted alkene precursors is illustrated in Fig. 8. Due to the absence of a conjugated acryl group, the sulfur-substituted alkene (species **9**) herein is relatively inert to the addition of radicals compared to aromatic monomers. Thus, the sulfur radicals participate in the H-abstraction of α-methylene moieties near the alkenyl group, leading to a conjugated di-alkene substituted by a sulfur chain. Then, the sulfur chain breaks into sulfur radicals under heat and attacks the remote alkenyl group (Fig. 8). A five-membered ring intermediate consequently comes into being, and the further H-abstraction makes it transform into the thiophene moiety finally. The transformation mechanism involving a conjugated diene moiety is further

corroborated by the degradation of SA. The inverse vulcanization of SA generated 5-methylthiophene-2-carboxylic acid in large quantities (Supplementary Fig. 44), demonstrating that the conjugated diene moiety favors the formation of the thiophene domain.

DFT calculations were carried out to illustrate the evolution of sulfur-substituted alkenes into thiophene. The olefin-like species **9** was first extracted with a proton radical via **TS₉₁₀** with $\Delta G_{910} = 10.2$ kcal/mol, and the repeating extraction towards stable conjugated species **10** with low energy (−14.1 kcal/mol) occurred spontaneously. Under the heat, species **10** was confirmed to undergo a two-step cyclization in a radical manner with $\Delta G_{1011} = 36.4$ kcal/mol, leaving proton radical eventually to generate stable aromatic thiophene species **11** (−32.5 kcal/mol).

In sum, with the help of DFT calculation and the identification of intermediates and by-products, we demonstrated that in classical inverse vulcanization, the monomers composed of alkenyl group conjugated with aryl groups are prone to degrade via sulfur-substituted alkenes into species containing thiocarbonyl moieties, especially DTT which is an aromatic and stable group. In comparison, without the activation and methyl end group for rearrangement, the isolated alkenyl moieties located at the middle of the olefin chains are difficult to degrade into thiocarbonyls. Instead, the long hydrocarbon chains provide opportunities for the sulfur-substituted alkene intermediates to cyclize into thiophene moieties at relatively slow rates (Fig. 9).

The distinct mechanisms clarify the difference in inverse vulcanization curves of various crosslinker monomers. As discussed above, similar to the classical vulcanization curve of rubbers crosslinked by elemental sulfur, for the inverse vulcanizations using a multifunctional alkene as a crosslinker, a distinct inverse vulcanization curve can also be established based on the moduli variation with time during the reaction. The inverse vulcanization process can be generally divided into periods, including the induction stage, curing stage, and over-cure stage after the optimum cure stage (i.e., moduli maximum or plateau). In the induction period, the moduli are nearly unmeasurable because of the low monomer conversion; in the curing stage, the moduli undergo a rapid increment towards a maximum or plateau as

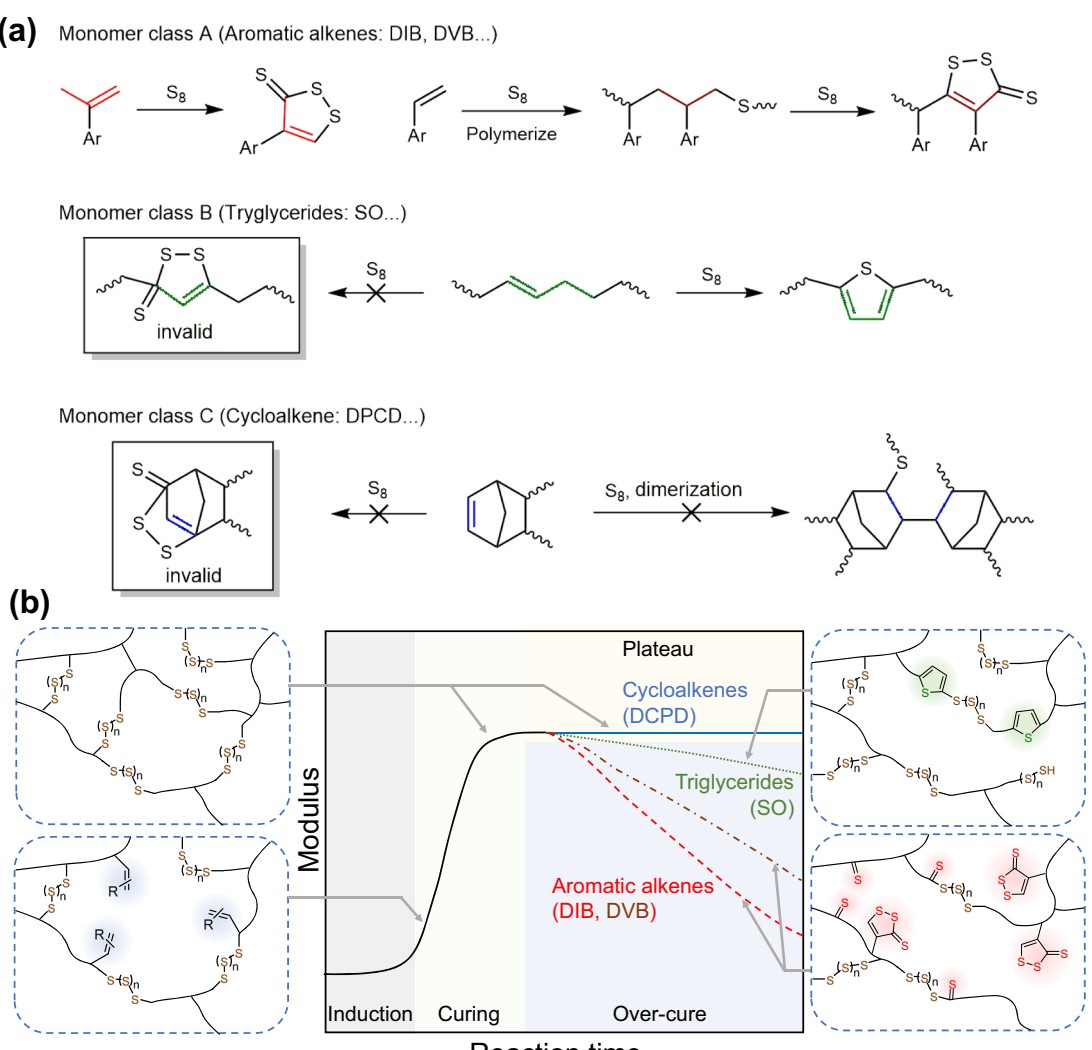

**Fig. 9 | Side reactions and inverse vulcanization curves depending on monomers. a** The possible side reactions and the by-products in the inverse vulcanization of various classes of monomers at high temperatures (>160 °C). **b** The patterns of the inverse vulcanization curve of various monomers with the network and segment structures illustrated at different stages.

long chains of polysulfides are growing and networked via organic monomers with crosslinking and/or branching points being increasingly formed towards a saturated state. After the optimum cure point, the pattern of the over-cure stage is largely dependent on the monomer structure. In the inverse vulcanization of DIB, the isopropenyl group conjugated to the phenyl group is the easiest structure to experience H-abstraction followed by rearrangement into DTT (Fig. 9a). Therefore, poly(DIB-S) degrades in very fast manners regardless of the temperature and the sulfur feed ratio. The degradation breaks the organic sulfide moieties in linear units and branch points into stable DTT dead ends, therefore reducing the crosslinking densities and increasing the $M_c$ (Table 1 and Fig. 9b). The $G'$ of the products consequently declines with accelerated relaxation of the network due to the plasticization effect of free ends (Fig. 2). On the other hand, due to the lack of methyl substitution on the alkenyl group in DVB monomer for the formation of CDTA and DTT moieties, the degradation of sulfide units in DVB requires dimer microstructures (Fig. 9a). Hence, the inverse vulcanization product of DVB deteriorates in a relatively slow manner and needs higher temperatures (Supplementary Figs. 17, 18). In comparison, due to the absence of an activation group and alkenyl end in SO, the sulfur-substituted alkene generated from H-abstraction in SOS, i.e., poly(SO-S) can only cyclize into thiophene slowly (Fig. 9a). Since the formation of thiophene

breaking the branch points is slow and generates few pendant ends, there is a minor drop in $G'$ of the SOS without significant alteration in relaxation behaviors (Figs. 3 and 9b). Conversely, the accumulation of rigid and aromatic thiophene species in the products retard the relaxation behaviors, resulting in the abnormal increase in $G'$, $E_a$, and $\tau_n$ (Fig. 3a and c). The slow deterioration also explains why the over-cure stage is only observed at high reaction temperatures in inverse vulcanization of SO. Lastly, because the cycloalkenes in DCPD can only self-polymerize following the metathesis mechanism in the presence of metal catalysts[62], the cycloalkene structure restricts not only the rearrangement of sulfur-substituted alkene into thiocarbonyls but also its cyclization into thiophenes due to the lack of flexible alkenyl chains or dimerization (Fig. 9a). Therefore, the inverse vulcanization of DCPD exhibits superior stability towards high reaction temperatures with a lasting plateau stage with high modulus (Supplementary Figs. 29–32 and Fig. 9b).

## Methods
### Chemicals

1,3-Diisopropenylbenzene (DIB, >97%, Aladdin, China), 1,2-divinylbenzene (DVB, 80%, Energy Chemicals, China), allyl phenyl ether (APE, 97%, Aladdin, China), α-methylstyrene (αMSt, 96%, Aladdin, China), 4-methylstyrene (4-MSt, 96%, Aladdin, China), soybean oil (SO,

Yihai Kerry Arawana Holdings Co, Ltd), EOD (Adamas-beta, China), sorbic acid (SA, >99%, Adamas-beta, China), DCPD (98%, Adamas-beta, China), and elemental sulfur ($S_8$, sublimed, 99.5%, Sinopharm, China) were used as received.

## Characterization

**Nuclear magnetic resonance.** NMR spectra including, $^1H$, $^{13}C$, $^1H$-$^1H$ correlation spectroscopy ($^1H$-$^1H$ COSY), heteronuclear multiple bond correlation (HMBC), and heteronuclear single quantum coherence spectroscopy (HSQC) spectra, were acquired on a Bruker AV 400 M spectrometer. $CDCl_3$ was used as the solvent, with tetramethylsilane as the internal reference.

**Fourier transform infrared.** The IR absorption spectra of the samples were recorded on a Nicolet IS50 Fourier transform infrared spectrometer (Thermo Scientific). The KBr tablet method was applied to solid samples, which were tested in the transmission mode, while liquid or viscous samples were tested in the attenuated total reflection (ATR) mode. Every sample was scanned 32 times with a resolution of 4 cm$^{-1}$ in the range of 400 cm$^{-1}$ to 4000 cm$^{-1}$ and the average infrared signal was adopted.

**Ultraviolet and visible spectrophotometry (UV-Vis).** The UV-Vis absorption spectra of the samples were recorded on a TU-1901 UV-Vis Spectrophotometer.

**Differential scanning calorimetry (DSC).** DSC curves of samples were recorded on a DSC3 STAR differential scanning calorimeter (Mettler-Toledo, Switzerland) under a nitrogen atmosphere. About 5–10 mg sample was encapsulated in an aluminum-enclosed pot, before being first heated from 25 °C to 140 °C at a heating rate of 10 °C/min, and then to −50 °C at a cooling rate of 10 °C/min, and again to 140 °C at a heating rate of 10 °C/min. The DSC curve in the second heating cycle is presented for the determination of glass transition temperatures ($T_g$).

**X-ray photoelectron spectroscopy.** The sample chemical bonds were analyzed using an ESCANLAB Xi$^+$ XPS system with a monochromatic Al $K\alpha$ X-ray source. An electron/ion gun was used to compensate for charge build-up on the sample during measurement. The XPS peaks were fitted using the CasaXPS software, and the binding energy was corrected with a reference to C 1$s$ at 284.5 eV.

**Kinetic investigation.** Kinetic investigations into inverse vulcanization took monofunctional alkenes as model comonomers. As a typical polymerization for kinetic investigation, αMSt (1.30 g, 11.0 mmol) was mixed with $S_8$ (0.70 g, 21.8 mmol regarding sulfur atom) in a vial at room temperature. The mixture was then heated to 160 °C by an oil bath under stirring. Subsequently, ~0.2 mL samples were withdrawn from the liquid mixture at a pre-determined time (0.5, 1, 1.5, 2, 2.5, 3, 3.5, 4 h). The samples were subjected to NMR tests to determine the composition of the reaction mixture during inverse vulcanization.

**Rheology.** Rheological properties of the polysulfides synthesized from inverse vulcanization were measured by an HR-20 stress-controlled rheometer (TA measurement, USA) with a disposable 25 mm diameter parallel plate. Multi-functional alkenes were adopted as model cross-linkers in inverse vulcanization to prepare samples for rheological measurement.

**Time sweep.** As a typical time sweep test, DIB (1.10 g, 7.0 mmol) was mixed with $S_8$ (0.90 g, 28.1 mmol) in a vial at room temperature. The mixture was then heated to 160 °C by an oil bath under stirring. As the mixture turned into a homogeneous brown opaque liquid, about 0.5 mL of the mixture was rapidly transferred onto the mounted parallel plate pre-heated to 160 °C in the rheometer by a glass dropper.

The time sweep test was performed at an oscillation frequency ($\omega$) 2 rad/s at 160 °C with a strain amplitude ($\gamma$) of 0.2% for 4 h.

**Strain sweep and frequency sweep.** As a typical procedure for a strain sweep test followed by a frequency sweep, the sample prepared from inverse vulcanization was heat-pressed into a 1.2 mm disk under 10 MPa at 160 °C for 5 min. Then disk sample was mounted on the parallel plate pre-heated to 110 °C. A strain sweep test was first carried out from 0.01% to 200% at an oscillation frequency of 10 and 100 rad/s. According to the strain sweep results, frequency sweep tests were performed within the LVE regime with a strain amplitude of 0.2% from 100 rad/s to 0.05 rad/s at varied temperatures, respectively (110, 120, 130, 140, 150, and 160 °C in sequence).

**Soluble fraction test and composition analysis.** In a pre-weighed vial, a weighed sample ($m_O$) was immersed in 5 mL THF. After shaking at room temperature (25 °C) for 24 h, the solution fraction and the solid residue were separated. The solution was concentrated under vacuum for further analysis, including NMR investigation, while the solid residue was dried in an oven at 60 °C until a constant weight $m_p$, by which the soluble fraction was estimated to be $1-m_p/m_O$.

**Density.** The density of the sample was measured by an AE224J density balance (Sunny Hengping Instrument, China) at room temperature (25 °C) according to the buoyancy method using deionized water as the medium. A piece of the sample was weighed by the balance in air ($m_1$) and in water ($m_2$). The density of the sample can be calculated from $m_1$, $m_2$, and the density of water ($\rho = 1.000$ g/cm$^3$) by $\rho \times m_1/(m_1-m_2)$. Every sample was tested 8 times and the three highest values were averaged as the measured density of the sample.

**Density functional theory calculation.** DFT calculations were introduced to demonstrate the mechanism of the reaction process among various species. All geometries, including intermediates and TSs, were optimized under M06-2X/6-311 + G(d,p) level with tight criteria using an ultrafine grid, to ensure the accuracy of sulfur atoms. Frequencies of intermediates and TSs were checked to confirm that they had zero and one imaginary frequency, respectively, and intrinsic reaction coordinate was applied to investigate the neighboring reaction pathways of all TSs. Gaussian 16 program (Rev. B.01.) was employed to perform calculations[63].

**Element analysis (CHNS).** The elemental content was determined by a Vario EL Cube elemental analyzer (Elementar Analysensysteme GmbH, Germany). The combustion temperature was 1000 °C.

## Data availability
All data necessary to support the conclusions of this paper are available within the paper and in the Supplementary Information, including detailed material synthesis and characterization. In particular, the coordinates data of the optimized structures generated in this study are provided in the Source Data file, and the rheological data generated in this study are provided in the Supplementary Information. All other data that support the findings of this study are available from the corresponding authors upon request. Source data are provided in this paper.

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

## Acknowledgements
B.Z. acknowledges the financial support for this work from the Natural Science Foundation of Fujian Province of China (2022J05041). H.Z. acknowledges the support from the National Natural Science Foundation of China (22172028 and 21903015) and the Award Program of Fujian Minjiang Scholar Professorship (2018). T.B. acknowledges the support from the National Natural Science Foundation of China (22201105). H.S. acknowledges the Natural Science Foundation of Fujian Province of China (2022J011116).

## Author contributions
B.Z. and H.Z. conceived the project and designed the experimental procedures. L.Z., X.W., and P.L. carried out the chemistry experiments, and L.Z. and Z.Y. performed the rheological tests. B.Z. and H.S. analyzed the experiment data. H.S. and H.Z. supervised the execution of the experiments and data analysis. T.B. performed the DFT calculation to elucidate the mechanism. B.Z. and L.Z. drafted the paper, which was polished by H.Z. and H.S., and all authors reviewed the final manuscript.

## Competing interests
The authors declare no competing interest.
