## [Peer Review File · Nature Communications]

Reviewers' Comments:

Reviewer #1:

Remarks to the Author:

The manuscript by Zheng et al. reports on rheological and computational chemistry studies on the mechanism of the inverse vulcanization process with styrenic and unsaturated triglyceride ester monomers. The manuscript is divided into a rheological study of the curing process and high temperature reflow processes of these sulfur polymers, followed by DFT calculations of model compound alkenes to infer on polymerization mechanistic processes. Overall the work is done well and with sufficient detail, so the work is certainly important to warrant publication in a more focused journal. But this submission is deemed not sufficiently novel for Nature Communications based on the lack of truly new, or distinctive findings than the previous reports. The reported studies largely agree with the previous reports (Bao et al., JACS 2023, 14, 7553) on the mechanistic aspects of inverse vulcanization, and do not present new or distinctive rheological findings vs an earlier report (Bischoff et al. Nature Comm, 2023, 14, 7553). Hence, the work is recommended for publication elsewhere.

Reviewer #2:

Remarks to the Author:

This paper compares the progress of inverse vulcanization through evolving dynamic viscoelastic moduli of polymers prepared from three different types of monomers with sulfur, mostly at sulfur to monomer mass ratios of 2 to 1 and some at 4 to 1. They conclude from time sweeps that the networks from aromatic alkene monomers degrade quickly with time while the networks from cycloalkenes degrade much more slowly time. They then suggest reaction mechanisms behind this different behavior that they term overcure.

The authors have not discussed dynamic exchange reactions in the entire paper. Is there any stage or temperature where dynamic exchange reactions occur? This can be answered by analyzing the dynamic moduli obtained at various temperatures and presented as frequency response curves in the Supplemental information. This paper would be more connected to recent literature on rheology of inverse vulcanized polymers if this aspect were discussed.

Reviewer #3:

Remarks to the Author:

The manuscript describes a study of the reaction process of inverse vulcanisation. The authors (accurately) identify that there is currently a lack of sufficient understanding of this process in the literature. They use rheology to study this process at various stages of the reaction – typically in the literature these polymers are often made by a reaction stage followed by a curing stage. The results of this work suggest that inverse vulcanisation follows three stages – induction, curing, and over-cure. The authors point out that this is like the behaviour of conventionally vulcanised rubbers. This is perhaps unsurprising, as the processes are clearly closely related. However, these results will no doubt be useful and interesting to all researchers in the field of inverse vulcanisation. The discussion of the various potential degradation routes, and figure 6, is excellent work. An interesting finding is the densification of the polymers over extended curing – something I have not seem reported previously. I think this is a thorough, useful, and important study. I'd be very happy to see it published. The paper is generally well written, with the prior background, motivation, and importance well set out and referenced. The discussions are well supported by the results. At times the language could be clearer. I would recommend the authors to go through the paper and amend the language to try to keep it simple, clear, and scientific. The title: "Inverse Vulcanization Curve: Unveiling the Universality and Complexity of Structural Evolution during Inverse Vulcanization via Rheology and Mechanism Investigation of Different Monomers" – this does not need to be half so long, the abstract will explain

the details. The title could simply be "Structural evolution during inverse vulcanisation" (or similar, according to the authors choice). The authors give glowing reports of the previous research efforts, variously describing them as "masterpieces", "excellently describing", "enormous efforts", "pioneering work" etc. Its great to see the positivity, and credit being given to other researchers, but also feels a bit subjective – might be better to keep it dry and scientific, but that's a personal choice, I think.

Minor corrections:

1. The authors state ". To exclude the scale effect, all the inverse vulcanizations were carried out with the same total weight (2.0 g) of feedstocks" This makes perfect sense to me. In my own lab, we have found that reactions can behave very differently at different scale. The reaction is quite exothermic and insulating. So, behaves very differently at different scales, because of changes in heat loss. This is a significant effect, and potential complication in terms of scale up. For readers new to this area, the reasoning behind the authors statement about the scale might not be apparent – I think it could be worth them adding some discussion of what they meant by it, and why.
2. "In typical vulcanization of rubber by sulfur, the vulcanization process monitored by a vulcameter" – many readers, myself included, might not be familiar with a vulcameter -please could you add some explanation?
3. "This stage can be denoted as an induction/scorch period" – scorch is a commonly understood term in the rubber industry, but not in academic chemistry – please explain.
4. "Sbondnds exchange"? Typo? S-S bond exchange?
5. Interesting to see that the S-DCPD did not suffer the same loss of G' and solubility at extended curing. The discussion later is excellent, in terms of the various degradation routes. But could it also be related to a shorter sulfur rank? Could you check with XPS as you did for the DIB?
6. Figure 5, could you add some sort of legend or key showing the structures referred to visually? I think it would make this easier to absorb for the reader.
7. Soluble fraction method – the methods section says DCM, but the main text says THF – please provide clarity here.
8. If you have access, could you run elemental analysis (CHNS) of the samples at the different reaction stages? This would show any changes to the sulfur proportion over the reaction, as well as any changes to the C/H ratio, which would indicate H₂S loss.

Reviewer #4:

Remarks to the Author:

I have read this manuscript "Inverse Vulcanization Curve: Unveiling the Universality and Complexity of Structural Evolution during Inverse Vulcanization via Rheology and Mechanism Investigation of Different Monomers" with much interest. The authors present a thorough experimental investigation of how the mechanical properties of inverse vulcanization networks change upon varying the degree of curing, i.e. upon varying the inverse vulcanization time. The rheological tests provide a wealth of information and details into the network relaxation and viscoelastic properties which are then qualitatively interpreted in terms of the molecular-level structure and underlying chemical reactions. The main finding is that the the mechanical properties vary in a non-monotonic way as a function of the inverse vulcanization time, with over-curing corresponding to degradation of the mechanical response. The manuscript is well written and the results are scientifically sound, novel and surely of great interest for a broad audience of chemists, physicists and material scientists. Before making a final recommendation I would like the authors to address the issue described below.

- The investigation is very detailed under the chemical reaction and rheological experimental profiles. What is lacking is a more quantitative physical mechanism to link the molecular level chemistry and structure of the network to the resulting mechanical properties. For example, the authors could try and link the variation of mechanical glass transition temperature to the molecular-level degree of covalent bonding using the model of Phys. Rev. Lett. 110, 178002 (2013), by keeping in mind the role of coordination number of the covalent bonds on the elastic modulus of the network, Modern Physics Letters B Vol. 27, No. 05, 1330002 (2013).

- in Fig. 3(b) the dynamical curves reveal a significant increase of the modulus with frequency of oscillation, which is indicative of nonaffine displacements and nonaffine elasticity (see above and *Soft Matter* 14, 8475-8482 (2018)).

- the title is very long, the authors could consider a shorter version.

Responses to Reviewers' Comments

Inverse Vulcanization Curve: Unveiling the Universality and Complexity of Structural Evolution during Inverse Vulcanization via Rheology and Mechanism Investigation of Different Monomers

We would like to thank the reviewers for their thorough reviews of our manuscript and providing excellent and constructive comments. We revised our manuscript to address all the concerns raised. Below are the itemized responses in blue to each comment in *italics*. The revisions which were made in the manuscript to address the relevant comments were highlighted in yellow and copied below to facilitate the reading.

REVIEWER COMMENTS

Reviewer #1 (Remarks to the Author):

The manuscript by Zheng et al. reports on rheological and computational chemistry studies on the mechanism of the inverse vulcanization process with styrenic and unsaturated triglyceride ester monomers. The manuscript is divided into a rheological study of the curing process and high temperature reflow processes of these sulfur polymers, followed by DFT calculations of model compound alkenes to infer on polymerization mechanistic processes. Overall the work is done well and with sufficient detail, so the work is certainly important to warrant publication in a more focused journal. But this submission is deemed not sufficiently novel for Nature Communications based on the lack of truly new, or distinctive findings than the previous reports. The reported studies largely agree with the previous reports (Bao et al., JACS 2023, 14, 7553) on the mechanistic aspects of inverse vulcanization, and do not present new or distinctive rheological findings vs an earlier report (Bischoff et al. Nature Comm, 2023, 14, 7553). Hence, the work is recommended for publication elsewhere.

Thanks for the interesting comment and your time given in reviewing our manuscript is greatly appreciated. The references you mentioned indeed raised important findings about the structures and rheological behaviors of inverse vulcanization products, respectively. However, we would like to emphasize that our manuscript is focused on a totally distinct topic, i.e., the structural evolution of products during inverse vulcanization, and the similarity between our manuscript and the mentioned literature, if have, is limited in the usage of some similar investigation methods.

The first reference (Bao et al, J. Am. Chem. Soc. 2023, 14, 7553, if you mean Bao et al, J. Am. Chem. Soc. 2023, 145, 22, 12386) revealed that the cumyl moieties is the major products of the inverse vulcanization of DIB. The second reference (Bischoff et al., Nat. Comm. 2023, 14, 7553) investigated the rheological properties of inverse vulcanization products of DIB, styrene

and dimeric norbornadiene and their copolymers. They found that the relaxation modes of adaptable network in terms of Maxwellian behaviors varied upon altering the composition of alkenyl crosslinkers. Nevertheless, although systematic and profound research presented, both of them neglected the evolution of the inverse vulcanization products before and after the vitrification but used just vitrified polysulfides as model polymers as stated in the experimental parts of the references. It is noted that the reaction time is undoubtedly critical to the structures and properties of the inverse vulcanization product considering potential side reactions due to the high temperature, which greatly influence the properties of polysulfides as observed in few references without explanations.

Therefore, to establish guideline for development of high-performance polysulfides with well-defined structure, in our manuscript, we focused on the structural and property evolution of inverse vulcanization products with reaction time instead of assuming inalterable structures after vitrification as the references did. Combining rheology methods, NMR characterizations and DFT calculation, for the first time we systematically revealed the structural evolution of inverse vulcanization products depending on alkene structures, i.e. aromatic alkenes, olefins and cycloalkenes, and a few major side product moieties including thiocarbonyls and thiophenes are identified with generation mechanism elucidated and confirmed by intermediates, which has never been reported. The structure evolution and its dependence as well as mechanisms revealed throw light upon the eccentric and interesting phenomena reported, such as the products with diverse properties generated from the same crosslinker but different reaction conditions (Polym. Chem. 2023, 14, 943; Macromolecules 2000, 33, 9353; ACS Macro Lett. 2015, 4, 9, 862), and the enforcement effect of DCPD (Chem. Eur. J. 2019, 25, 10433). Furthermore, the study contributes to the understanding of basic inverse vulcanization process and structure-reaction condition relationship, which is essential for optimization of inverse vulcanization conditions for various alkenes to precisely synthesize polysulfide in developing high-performance sulfur-rich materials.

Along with the other three reviewers that appreciated the topic and the novelty of this work, we do think the study can be of great interest to the broad readership of *Nature Communications*. Therefore, we sincerely request for an opportunity from you to kindly reconsider this work.

To highlight the importance and novelty of our work, we reorganized the introduction in the revised manuscript as follows:

Page 4 in the revised manuscript (the deleted sentences and words are not shown to ensure the readability):

..... a complete understanding of inverse vulcanization mechanisms and product structures is coming to light.

However, although there is a lot of research focusing on determining the properties and unit structures of inverse vulcanization products prepared from given conditions assuming a fixed structures, and the evolution of the polysulfide network and segment structures during inverse vulcanization-as well as its mechanism is still elusive. Among rare reports, J. Griebel found that in the inverse vulcanization of DIB, a too-long reaction time at high temperatures would lead to a

flow transition of the polymer product, suggesting the breaking of the polymer network²⁹. In contrast, D. Kim and his coworkers³⁰ discovered that the post-thermal treatment at 110 and 140 ° will improve the thermomechanical properties of inverse vulcanization products based on divinylbenzene in terms of glass transition temperature (T_g) and storage modulus. C. Jenkins et al.³¹ investigated the effects of different reaction conditions including reaction time, temperatures, scales, and feeding and heating procedures. The results showed that all the factors would influence the gel fractions, the glass transition temperature and the molecular weight of soluble fractions, which are closely related to degree of crosslinking. The complex relationship between reaction condition and product properties observed clearly indicates that the reaction conditions and procedures adopted in different research greatly influence the structures and structural evolution of inverse vulcanization products, which dictates the network and segment behaviors of ultimate products. Nevertheless, only preliminary explanations were provided to elucidate the evolution of product structures and behaviors. The degradation of polysulfides at long reaction time was usually intuitively ascribed to the break and rearrangement of sulfur chains^{28, 30}. Moreover, most mechanism investigation and thermal treatment investigations were carried out with sole model monomer^{20, 21}. It is not sure whether the mechanism can be extended to other monomers, and the monomer dependence of the structural evolution is also obscure. For the reason of lacking knowledge about the structural and property evolution as well as its relation with polymerization conditions and the alkene structures, the precise synthesis of polysulfides with controlled structures by inverse vulcanization is still challenging, which perturbs the development of polysulfide materials. Therefore, it is of great necessity to understand the evolution of inverse vulcanization products and the mechanism behind based on diverse alkenyl comonomers in various synthesis conditions.

In this work ...

Reviewer #2 (Remarks to the Author):

This paper compares the progress of inverse vulcanization through evolving dynamic viscoelastic moduli of polymers prepared from three different types of monomers with sulfur, mostly at sulfur to monomer mass ratios of 2 to 1 and some at 4 to 1. They conclude from time sweeps that the networks from aromatic alkene monomers degrade quickly with time while the networks from cycloalkenes degrade much more slowly time. They then suggest reaction mechanisms behind this different behavior that they term overcure.

The authors have not discussed dynamic exchange reactions in the entire paper. Is there any stage or temperature where dynamic exchange reactions occur? This can be answered by analyzing the dynamic moduli obtained at various temperatures and presented as frequency response curves in the Supplemental information. This paper would be more connected to recent literature on rheology of inverse vulcanized polymers if this aspect were discussed.

Thank you for the constructive comments. According to the advice, we analyze the temperature where dynamic exchange reactions occur along with the dynamic mechanism based on modulus-frequency curves.

It should be kept in mind that the S-S bond exchange is a chemical reaction whose rate depends on temperatures, and even at room temperature the exchange reaction happens imperceptibly. Hence, the activation temperature for dynamic exchange is supposed to be defined as the temperature where the relaxation of network can be achieved via S-S bond exchange, i.e., $G'' > G'$, at a given time-scale. According to the $G-\omega$ curves of poly(DIB-*r*-S) obtained from temperature ramps. (Supplementary Figs. 6-9), the crossover of G' and G'' is observed in $G-\omega$ curves of all the samples obtained at 120 °C. In contrast, for poly(SO-*r*-S), the crossover of G' and G'' firstly appear in $G-\omega$ curves obtained at 130 °C and no crossover is captured at 120 °C. The results reveal that, within the timescale of measurement, i.e. shorter than ~ 120 s (0.05 rad/s) which is comparable to the typical timescale of polymer processing, the dynamic exchange is regarded to be observable at about 120 °C depending on the crosslinkers and inverse vulcanization conditions.

We also tried to retrieve the activation temperature and mechanism of the S-S bond exchange from vertical shift factors (b_T), i.e., the ratio between G_N^0 at the measurement temperature and that at the reference temperature, as did in the reference (Nat. Commun. 2023, 14, 7553). In the frequency sweep results and b_T plots of poly(DIB-*r*-S) (Supplementary Figs. 6-10) and poly(SO-*r*-S) (Supplementary Figs. 23-27), the plateau modulus of the polymer (represented by b_T) increase as the measurement temperature was increased except for the poly(DIB-*r*-S) samples measured at 160 °C. The experimental b_T are in consistency with (DIBS) or higher than (SOS) theoretical b_T dictated by entropy elasticity (grey line in Supplementary Fig. 10) as equation 2 in the manuscript and Supplementary equation 1 expressed, which indicates the trace loss of crosslink density. The trend implies that the relaxation via S-S bond exchange majorly follows associate mechanism as disulfide-crosslinked materials reported (Mater. Horiz., 2014, 1, 237-240; Macromolecules 2020, 53, 1884), which is different from dissociative mechanism claimed in the Nat. Commun. 2023, 14, 7553 wherein inverse vulcanization conditions and feed ratios of sulfur differed from our samples. As the temperature was increased to 160 °C, the loss of crosslinks is observed in poly(DIB-*r*-S), suggesting the generation of dissociative sulfur radical species. Anyway, due to the dominant associate mechanism, the activation temperature cannot be directly inferred from b_T herein.

The discussion is added to the revised manuscript and Supplementary Information as follows:

Page 10 in the revised manuscript:

...adaptable covalent networking microstructure has a strong correlation with temperature (frequency). The crossover of G' and G'' is observed in $G-\omega$ curves of all samples (Supplementary Figs. 6–9) measured at 120 °C and higher temperatures, but does not exist in the curves at 110 °C. This indicates that, within the timescale of measurement, i.e., shorter than ~ 120 s (0.05 rad/s) which is comparable to the typical timescale of polymer processing, the dynamic exchange is

regarded to be active at 110–120 °C. In low temperatures (110 °C) below the activation temperature of the adaptable covalent bonds (or a time shorter than its characteristic relaxation time, presumably corresponding to the G'-G'' crossover), the network microstructure is presented (i.e., $G' > G''$), while above the activation temperature (or a time longer than its characteristic relaxation time) the rheological response is suggestive of dissociated-linear polymer (i.e., $G' < G''$).

The dynamic exchange mechanism of S-S can be inferred from vertical shift factors (b_T , Supplementary Fig. 10) in the superposition, i.e., the ratio between G_N^0 at the measurement temperature and that at the reference temperature^{26,38}. In the frequency sweep results and b_T plots (Supplementary Figs. 6–10), the plateau modulus increased (represented by b_T) as the measurement temperature was increased except for the DIBS samples measured at 160 °C. The experimental b_T are in consistency with theoretical b_T dictated by entropy elasticity as Supplementary equation 1 expresses, which indicates the trace loss of crosslink density. The increase trend implies that the relaxation via S-S bond exchange majorly follows associate mechanism reported in disulfide-crosslinked materials³⁹. As the temperature was increased to 160 °C, the modulus contrarily declined probably as a result of the loss of crosslinks, indicating the presence of sulfur radical species and dissociate exchange mechanism²⁶.

To discern the dynamics...

Page 17 in the revised manuscript:

...and the weighted relaxation spectra are thereby plotted (Fig. 3c). The activation temperature of dynamic exchange reaction mediated by polysulfur bond is estimated to be 120–130 °C on basis of appearance of G'-G'' crossover in the G- ω curves measured at 130 °C rather than 120 °C (Supplementary Figs. 23–26). The monotonous increase of b_T with temperatures (Supplementary Fig. 27) demonstrate the associate mechanism is dominant in the relaxation of dynamic SOS via S-S bonds.

As shown in Fig. 3b...

Reviewer #3 (Remarks to the Author):

The manuscript describes a study of the reaction process of inverse vulcanisation. The authors (accurately) identify that there is currently a lack of sufficient understanding of this process in the literature. They use rheology to study this process at various stages of the reaction – typically in the literature these polymers are often made by a reaction stage followed by a curing stage. The results of this work suggest that inverse vulcanisation follows three stages – induction, curing, and over-cure. The authors point out that this is like the behaviour of conventionally vulcanised rubbers. This is perhaps unsurprising, as the processes are clearly closely related. However, these results will no doubt be useful and interesting to all researchers in the field of inverse vulcanisation. The discussion of the various potential degradation routes, and figure 6, is excellent

work. An interesting finding is the densification of the polymers over extended curing – something I have not seem reported previously. I think this is a thorough, useful, and important study. I'd be very happy to see it published.

The paper is generally well written, with the prior background, motivation, and importance well set out and referenced. The discussions are well supported by the results. At times the language could be clearer. I would recommend the authors to go through the paper and amend the language to try to keep it simple, clear, and scientific. The title: “Inverse Vulcanization Curve: Unveiling the Universality and Complexity of Structural Evolution during Inverse Vulcanization via Rheology and Mechanism Investigation of Different Monomers” – this does not need to be half so long, the abstract will explain the details. The title could simply be “Structural evolution during inverse vulcanisation” (or similar, according to the authors choice). The authors give glowing reports of the previous research efforts, variously describing them as “masterpieces”, “excellently describing”, “enormous efforts”, “pioneering work” etc. Its great to see the positivity, and credit being given to other researchers, but also feels a bit subjective – might be better to keep it dry and scientific, but that's a personal choice, I think.

We express our great gratitude to you for your professional review on our manuscript and your kind appreciation of our work! According to your nice suggestions, modification has been made in the manuscript:

We agree that the title could be too long for the reader to grasp the essence of the paper. Hence, the title has been simplified to “Structural evolution during inverse vulcanization”. On the other hand, we have gone through the manuscript carefully, polished the expression and corrected the typos where necessary. The glowing words used to describe the previous research in the introduction part have been deleted or substituted with dry and scientific descriptions. Because of the extensive modifications in narration which is hard to be contained within the rebuttal letter, please see the revised manuscript for details.

Minor corrections:

1. The authors state “. To exclude the scale effect, all the inverse vulcanizations were carried out with the same total weight (2.0 g) of feedstocks” This makes perfect sense to me. In my own lab, we have found that reactions can behave very differently at different scale. The reaction is quite exothermic and insulating. So, behaves very differently at different scales, because of changes in heat loss. This is a significant effect, and potential complication in terms of scale up. For readers new to this area, the reasoning behind the authors statement about the scale might not be apparent – I think it could be worth them adding some discussion of what they meant by it, and why.

Thank you for this great advice! To help the readers to understand the reason behind the statement regarding reaction scale, we add explanation and discussion to the relevant sentences as follows:

Page 5 in the revised manuscript:

...has sparse alkenyl moieties to sequester sulfur. On the other hand, it has been demonstrated that the reaction scale significantly influences reaction behaviors and the properties of the inverse vulcanization products³¹ especially when the amount of feedstocks is big, because the high viscosity of the reaction mixtures in curing retards transfer of substances and heat released by exothermic polymerization. To exclude the disturbance of scale effect complicating the structural evolution, all the inverse vulcanizations were carried out with the same total weight (2.0 g) of feedstocks.

2. *“In typical vulcanization of rubber by sulfur, the vulcanization process monitored by a vulcameter” – many readers, myself included, might not be familiar with a vulcameter -please could you add some explanation?*

Thank you for the comment. ‘Vulcameter’ is a specialized rheometer to investigate the vulcanization of a rubber sample. The torque and other mechanical properties of the sample can be measured during the vulcanization to determine the time of scorch, curing and other stages. To avoid unclarity, we replace the ‘vulcameter’ with ‘rheometer’ in the revised manuscript.

Page 6 in the revised manuscript:

In typical vulcanization of rubber by sulfur, the vulcanization process monitored by a vulcameter rheometer could be divided into three stages: induction, curing and over-cure stages in sequence (Fig. S1),...

3. *“This stage can be denoted as an induction/scorch period” – scorch is a commonly understood term in the rubber industry, but not in academic chemistry – please explain.*

Thanks for the advice. In rubber manufacture, the scorch period is a short-term initial vulcanization stage after addition of the sulfur additives, during which the rubber polymer chains are either not crosslinked yet by the sulfur or only partially crosslinked, followed by a stage where the crosslinking reaction becomes dominant. In scorch period, the “vulcanized” rubber polymer chains still have good flow ability and can be readily processed at a given temperature, in contrast to the following curing stage wherein the rapid formation of crosslinking hinders the flow and easy processing of rubber. The explanation has been added to the revised manuscript.

Page 7 in the revised manuscript:

This stage can be denoted as an induction/scorch period akin to that induction/scorch period of classical rubber vulcanization, wherein not yet or partially crosslinked rubber polymer chains still can be readily processed at given temperatures before curing stage begins, as shown in a typical vulcanization curve (Supplementary Fig. 1).

4. *“Sbondnds exchange”? Typo? S-S bond exchange?*

Thanks for your careful checks. We are sorry for our carelessness. It has been corrected in the revised manuscript.

Page 11 in the revised manuscript:

Herein, the G'-G'' crossover is most likely the characteristic of dynamic covalent network relaxation realized by S-S bonds exchange.

5. Interesting to see that the S-DCPD did not suffer the same loss of G' and solubility at extended curing. The discussion later is excellent, in terms of the various degradation routes. But could it also be related to a shorter sulfur rank? Could you check with XPS as you did for the DIB?

Thanks for the advice. We check the XPS spectra (Fig R1) of inverse vulcanization product of DCPD with different reaction times and the sulfur ranks of the samples are calculated from deconvolution of S 2p signals. Interestingly, the poly(S-r-DCPD) has a sulfur rank of 7.66, 7.34, 7.12 and 7.30 at 80, 130, 180 and 240 min, respectively, which is higher than poly(S-r-DIB) and poly(S-r-SOS) (6-7). The high sulfur rank could be ascribed to incomplete conversion of alkenyl groups with steric hindrance which react with sulfur chains slowly as evidenced by the decreased sulfur rank as inverse vulcanization proceeded. Therefore, the trace loss of G' and gelation at extended curing is supposed to be ascribed to the nature of stability within DCPD instead of short sulfur chains unfavorable for side reaction. In addition, the declining sulfur proportion unveiled by elemental analysis (See response to 8th comment) confirmed that the generation of H₂S could be the only side reaction in inverse vulcanization of DCPD.

Fig. R1 The high-resolution S 2p XPS spectra of poly(DCPD-r-S) at different inverse

vulcanization time: (a) 80 min; (b) 130 min, (c) 180 min and (d) 240 min.

The XPS results is added to the Supplementary Information and referred in the revised manuscript:

Page 20 in the revised manuscript:

...coinciding with rheological measurement on verifying the negligible degradation. The trace degradation is also confirmed by XPS spectra and element analysis (Supplementary Figs. 12 and 33) that decreases in sulfur rank, sulfur proportion and hydrogen proportion are observed without the appearance of distinct sulfur moieties, indicating that the generation of H₂S is supposed to the only side reaction during inverse vulcanization of DCPD. That is, the inverse vulcanization...

6. Figure 5, could you add some sort of legend or key showing the structures referred to visually? I think it would make this easier to absorb for the reader.

Thanks for the nice suggestion to improve the readability of the figure. The colored structures are added following kinetic plots to explain what moieties the points refer to in Fig. 5.

Fig. 5 at Page 25 in the revised manuscript:

Fig. 5 The kinetic plot of αMSt inverse vulcanization under different reaction conditions....

7. Soluble fraction method – the methods section says DCM, but the main text says THF – please

provide clarity here.

Thanks for pointing out the ambiguity. In the experiments, we actually used THF as the solvent to determine the gel fraction thereby the lowest gel fractions can be obtained. Besides THF, a few solvents were tried in the experiments and 'DCM' wrote down in the old version manuscript is neglected to be substituted by 'THF' in 'Methods' section. To clarify the ambiguity, we corrected the description in 'Methods' section as follows:

Page 37 in the revised manuscript:

In a pre-weighed vial, a weighed sample (m_0) was immersed in 5 mL THF DCM. After shaking at room temperature....

8. If you have access, could you run elemental analysis (CHNS) of the samples at the different reaction stages? This would show any changes to the sulfur proportion over the reaction, as well as any changes to the C/H ratio, which would indicate H₂S loss.

Thanks for your constructive suggestion. We run elemental analysis (CHNS) on samples at different stages (DIBS1, SOS4 and poly(S-DCPD)). The C/H ratio and sulfur proportion is calculated and plotted against reaction time as Fig. R2. Theoretically, the generation of H₂S during inverse vulcanization would lead to a decrease in sulfur proportion and an increase in C/H ratio. As the figure shows, the sulfur proportion continually decline (48.1% to 47.2%) in DIBS as inverse vulcanization proceeds in contrast to the fluctuation of C/H ratio (10.55 to 10.53). The result suggests the release of sulfur-rich species including organosulfides besides H₂S during inverse vulcanization. The organosulfides generated from side reaction with relatively low boiling points evaporated from the reaction mixture leading to loss of carbon simultaneously, which can be seen as yellow oily substance sticking on the wall of the reaction vessel (Fig. R3a). The similar trend is observed in elemental analysis results of SOS4 with a drop in sulfur proportion (38.0% to 36.9%) albeit the obvious decrease of C/H ratio (7.038 to 6.7). This can also be rationalized by evaporation of organosulfide side products beside H₂S loss, especially considering the SO is a bio-derived mixture containing impurity with low boiling points (Fig. R3b). Lastly, the composition of inverse vulcanization products of DCPD experiences a drop in sulfur proportion (54.1% to 53.0%) accompanied with the increase in C/H ratio (10.0 to 11.1), demonstrating that the existence of H₂S loss but trace discharge of organosulfides in the stable DCPD-crosslinked polysulfides.

Fig. R2 Sulfur proportion and C/H ratio of inverse vulcanization products of (a) DIB, (b) SO and (c) DCPD presented at different reaction time as determined by element analysis (CHNS).

Fig R3. Photos of the reaction vials for inverse vulcanization of (a) DIB and (b) SO. The yellow/brown liquid is attached on the wall of the vials.

Anyway, the elemental analysis reveals that the loss of sulfur is prevalent in curing and over-cure stage. The loss of sulfur could result from both the release of both H_2S and organosulfide by-products, which requires further investigation.

The discussion is included into the Supplementary Information and is referred in the revised manuscript:

Page 14 in the revised manuscript:

...transformation of the sulfur chain into another sulfur-rich moiety. Interestingly, element analysis (CHNS) showed (Supplementary Fig. 12) that the sulfur proportion in DIBS declines (48.1% to 47.2%) slightly but continually during inverse vulcanization, which was usually argued to be ascribed to the generation of H₂S⁶, in contrast to the alteration of sulfur ranks. The result confirmed that besides the release of H₂S there are other side reactions involving sulfur chains.

The evolution of sulfur domains....

Page 19 in the revised manuscript:

...which suggests a different evolution mechanism in SOS. The sulfur proportion determined by element analysis (Supplementary Fig. 12) experiences a slight drop (38.0% to 36.9%) during inverse vulcanization similar to DIB, suggesting the limited generation of H₂S.

In sum,...

Page 20 in the revised manuscript:

...coinciding with rheological measurement on verifying the negligible degradation. The trace degradation is also confirmed by XPS spectra and element analysis (Supplementary Figs. 12 and 33) that decreases in sulfur rank, sulfur proportion and hydrogen proportion are observed without the appearance of distinct sulfur moieties, indicating that the generation of H₂S is supposed to be the only side reaction during inverse vulcanization of DCPD. That is, the inverse vulcanization...

Page 38 in the revised manuscript:

Element Analysis (CHNS). The elemental content was determined by a Vario EL Cube elemental analyzer (Elementar Analysensysteme GmbH, Germany). The combustion temperature was 1000 °C.

Thank you again for your positive and constructive comments and suggestions on our manuscript. We hope you will find our revised manuscript acceptable for publication.

Reviewer #4 (Remarks to the Author):

I have read this manuscript "Inverse Vulcanization Curve: Unveiling the Universality and Complexity of Structural Evolution during Inverse Vulcanization via Rheology and Mechanism Investigation of Different Monomers" with much interest. The authors present a thorough experimental investigation of how the mechanical properties of inverse vulcanization networks change upon varying the degree of curing, i.e. upon varying the inverse vulcanization time. The rheological tests provide a wealth of information and details into the network relaxation and viscoelastic properties which are then qualitatively interpreted in terms of the molecular-level structure and underlying chemical reactions. The main finding is that the mechanical properties vary in a non-monotonic way as a function of the inverse vulcanization time, with over-curing corresponding to degradation of the mechanical response. The manuscript is well

written and the results are scientifically sound, novel and surely of great interest for a broad audience of chemists, physicists and material scientists. Before making a final recommendation I would like the authors to address the issue described below.

- The investigation is very detailed under the chemical reaction and rheological experimental profiles. What is lacking is a more quantitative physical mechanism to link the molecular level chemistry and structure of the network to the resulting mechanical properties. For example, the authors could try and link the variation of mechanical glass transition temperature to the molecular-level degree of covalent bonding using the model of *Phys. Rev. Lett.* 110, 178002 (2013), by keeping in mind the role of coordination number of the covalent bonds on the elastic modulus of the network, *Modern Physics Letters B* Vol. 27, No. 05, 1330002 (2013).

Thanks for your constructive comment. We agree that the link the molecular level chemistry and network structure to the resulting macroscopic properties is valuable in understanding the system. Upon your advice, we tried to calculate the coordination number z_{co} of beads representing the polymer segment as follows:

According to *Phys. Rev. Lett.* 2023, 110, 178002. The z_{co} represents the covalent bond connected to the unit beads, which is to be calculated. The critical volume fraction ϕ_c , below which the nonaffinity deformation will significantly result in the loss of modulus, can be expressed by z_{co} as $\phi_c = \phi_c^* - \Lambda z_{co}$ where ϕ_c^* is the packing fraction in the limit $z_{co} = 0$. On the other hand, ϕ_c can be expressed as $\phi_c = \exp(-\alpha_T T_c - C)$, where α_T is thermal expansion coefficient and T_c is the critical temperature, i.e., glass transition temperature T_g in polymers. By expanding the exponential to linear order using Taylor formula and omit high order terms, one has $\phi_c = 1 - \alpha_T T_g - C$. With two equations involving ϕ_c , we obtain the relationship between T_c and z_{co} as follows:

$$\alpha_T T_g = 1 - C - \phi_c^* + z_{co} \Lambda \quad (R1)$$

Where $\Lambda \approx 0.1$, $C \approx 0.48$ pointed out by the literature considering similarity of DIB to styrene. For a random packing of hard spherical polymer beads, $\phi_c^* \approx 0.64$. Most polymers have a α_T ranging from 2×10^{-4} to $4 \times 10^{-4} \text{ K}^{-1}$ (Polymer, 1997, 38, 3485-3492) and an average of $3 \times 10^{-4} \text{ K}^{-1}$ is adopted in this study. By substituting the values into the equation, we have:

$$z_{co} = \frac{T_g}{3 \times 10^{-3}} + 1.2 \quad (R2)$$

Thereby, the z_{co} of samples DIB-S can be calculated from T_g , which is listed in column $z_{co,T}$ of Table R1:

Table R1 z_{co} of DIBS samples calculated from T_g and M_c .

Sample	T_g (K)	$z_{co,T}^a$	M_c (kDa)	$z_{co,M}^b$
DIBS1-1	283	2.049	93.2	2.011
DIBS1-2	308	2.124	13.4	2.075
DIBS1-3	312	2.136	14.8	2.067
DIBS1-4	309	2.127	47.9	2.021

^a As determined from T_g . ^b As determined from M_c .

The coordination number can also be estimated from molecular weight between crosslinks (M_c , Table R1). The DIBSs are crosslinked homogenous networks, therefore has infinite molecular weights. By assuming the molecular weight of Kuhn segment (M_k) as 1 kDa (J. Polym. Sci. B Polym. Phys, 2004, 42, 3505), the $z_{co,M}$ can be calculated from the number of Kuhn segments as beads between crosslinks ($n_k = M_c/M_k$) taking the coordination number of linear beads and crosslinked beads as 2 and 3, respectively:

$$z_{co,M} = \frac{2(n_k-1)+3}{n_k} \quad (R3)$$

The calculation result is showed in Table R1. Clearly the z_{co} calculated from T_g and M_c has similar trend of increase in curing followed by decrease in over-cure stage, although the discrepancy in values due to relatively arbitrary α_T and M_k used which surely differed as the unit structures evolved. Anyway, the calculation revealed that the decline in T_g in the over-cure stage is likely to arise from the loss of network connectivity.

As for poly(SO-*r*-S) (SOS), the parameters (Λ, C, M_k and α_T) of SOS samples which have entanglement and branch structures as well as unreacted alkenyl group simultaneously, should be distinct from those of polystyrene studied by the reference. Therefore, the z_{co} of SOS is hard to be estimated by the same algorithm, which requires further investigation.

The simplified discussion is added to the manuscript with the reference quoted, and detailed calculation is included in Supplementary Information:

Page 13 in the revised manuscript:

... induce plasticization effects. Moreover, by calculating network connectivity in terms of average coordination number (z_{co}) of polymer segments regarded as beads^{48, 49}(Supplementary Table 3 along with discussion), it is confirmed that the number of crosslink points increased followed by a decrease during inverse vulcanization, consistent with the change in M_c (Table 1).

To further ...

Page 41 in the revised manuscript:

48. Zaccane A, Terentjev EM. Disorder-assisted melting and the glass transition in amorphous solids. *Phys. Rev. Lett.* **110**, 178002 (2013).

49. Zaccane A. Elastic Deformations in Covalent Amorphous Solids. *Mod. Phys. Lett. B* **27**, (2013).

- in Fig. 3(b) the dynamical curves reveal a significant increase of the modulus with frequency of oscillation, which is indicative of nonaffine displacements and nonaffine elasticity (see above and *Soft Matter* 14, 8475-8482 (2018)).

We thank the reviewer for pointing out the reason behind the increase of modulus. An ascription to the increase of modulus with oscillation frequency to nonaffine displacements and nonaffine elasticity has been added to the revised manuscript with the reference cited as follows:

Page 17 in the revised manuscript:

In addition, the poor superposition due to the unexpected increase in modulus at high frequency ($> 10^1$ rad/s) is likely to result from nonaffine elasticity and nonaffine displacements of polymer segments as revealed in the reference⁵⁴.

Page 41 in the revised manuscript:

54. Palyulin VV, Ness C, Milkus R, Elder RM, Sirk TW, Zaccone A. Parameter-free predictions of the viscoelastic response of glassy polymers from non-affine lattice dynamics. *Soft Matter* 14, 8475-8482 (2018).

- the title is very long, the authors could consider a shorter version.

Thanks for the advice. We admit that the title was very long, and have shortened the title to “Structural evolution during inverse vulcanization” as suggested by the Reviewer #3 in the revised manuscript. We think the new title is concise for readers to grasp the essence of the manuscript.

Page 1 in the revised manuscript:

Structural evolution during inverse vulcanization

Reviewers' Comments:

Reviewer #2:

Remarks to the Author:

This revised manuscript addresses the question regarding the temperature range and description of dynamic bond exchange. Enough detail has been provided to make this manuscript worthy of publication.

Reviewer #3:

Remarks to the Author:

All of my comments have been addressed. Nice work. Happy to recommend publication with no further corrections.

Reviewer #4:

Remarks to the Author:

The authors have done an excellent job in addressing all the queries raised by the Referees. The manuscript has greatly improved thanks to the additional analyses and mechanistic insights after the revision process. I strongly recommend publication of this paper in Nature Communications as is.